# ADVERSARIAL SAMPLING FOR ACTIVE LEARNING

## ABSTRACT

This paper proposes ASAL, a new pool based active learning method that generates high entropy samples. Instead of directly annotating the synthetic samples, ASAL searches similar samples from the pool and includes them for training. Hence, the quality of new samples is high and annotations are reliable. ASAL is particularly suitable for large data sets because it achieves a better run-time complexity (sub-linear) for sample selection than traditional uncertainty sampling (linear). We present a comprehensive set of experiments on two data sets and show that ASAL outperforms similar methods and clearly exceeds the established baseline (random sampling). In the discussion section we analyze in which situations ASAL performs best and why it is sometimes hard to outperform random sample selection. To the best of our knowledge this is the first adversarial active learning technique that is applied for multiple class problems using deep convolutional classifiers and demonstrates superior performance than random sample selection[1].

## 1 INTRODUCTION

The goal of active learning (AL) algorithms is to train a model most efficiently, i.e. achieving the best performance with as few labelled samples as possible. Typical AL algorithms operate in an iterative fashion, where in each AL-cycle a query strategy selects samples that the oracle should annotate. These samples are expected to improve the model most effectively when added to the training set. This procedure continues until a predefined stopping criteria is met.

In this paper we will mainly focus on pool based active learning, because a pool of unlabelled samples is often available beforehand or can easily be build. Furthermore, annotating all pool samples serves as an ideal evaluation environment for active learning algorithms. It enables to train a fully-supervised model that establishes a performance upper bound on this data set. Similarly, randomly selecting instead of actively choosing samples establishes a lower bound. Then, the goal of an active learning algorithm is to approximate the performance of the fully supervised model with as few labelled samples as possible, while exceeding the performance of random sampling.

*Uncertainty sampling* is an effective query strategy that identifies samples that are more informative than random ones. The heuristic is, that samples for which the model is most uncertain contain new information and improve the model. However, to identify such samples an exhaustive search over the full pool is required and the uncertainty score needs to be recomputed as soon as the model is updated (each AL cycle). Thus, uncertainty sampling has a linear run-time complexity such that scanning very large unlabelled data sets is impractical even for inexpensive score functions.

Our contributions are as follows:

- We propose *Adversarial Sampling for Active Learning (ASAL)* that allows to approximate the performance of uncertainty sampling with a sub-linear run-time complexity.

- We conduct an extensive set of experiments using four different benchmarks (two and ten classes) and discuss the limitations of ASAL and how to overcome them.

- We demonstrate ASAL with different CNN based classifiers and three different feature sets to compare samples: raw pixel values, compressed representations of an auto-encoder and the features used to discriminate between real and fake samples in GANs.

---

[1]Code will be made publicly available upon publication.

## 2 RELATED WORK

We review related work on active learning especially on pool based uncertainty sampling and methods attempting to improve the run-time complexity of these active learning methods.

Pool-based active learning methods select new training samples from a predefined unlabelled data set (Hospedales et al. (2013); Nguyen & Smeulders (2004); Yang et al. (2015)). A common query strategy to identify new samples is uncertainty sampling( Joshi et al. (2009); Yang & Loog (2016)). Tong & Koller (2001) and Campbell et al. (2000) use minimum-distance sampling to train Support Vector Machines (SVMs). Minimum distance sampling is a well known uncertainty sampling strategy, it assumes that the classifier is uncertain about samples in the vicinity of the separating hyper-plane. This strategy is mainly used for two class but can be extended to multiple class problems by using the SVM in *one vs. one* or *one vs. all* settings (Jain et al. (2010)). Joshi et al. (2009) use information entropy to measure the uncertainty of the classifier for a particular sample. Computing uncertainty with information entropy is equally suitable for two or multiple classes.

Jain et al. (2010) propose two hashing based method to accelerate minimum distance sampling by selecting new samples in sub-linear time. These methods are designed to select the closest point (approximately) to a hyper-plane in a $k$-dimensional feature space, where the positions of the data points are fixed but the hyper-plane is allowed to move. Thus, these methods are limited to SVMs with fixed feature maps, because, if the feature map changes, the position of the samples become obsolete and need to be recomputed. Hence, the run time complexity is sub-linear for constant feature maps and linear otherwise. Unfortunately, CNN based methods update their feature maps during training. Thus, their methods are as efficient as exhaustive uncertainty sampling if CNNs are involved.

Zhu & Bento (2017) propose Generative Adversarial Active Learning (GAAL) that uses a Generative Adversarial Network (GAN), that is trained on the pool samples, to generate synthetic samples in each AL cycle. Generating instead of selecting uncertain samples leads to a constant run-time complexity because producing a new sample is independent of the pool size. Zhu & Bento (2017) use the traditional minimal distance optimization problem (see Eq. 1) but replace the variable $x$ (denoting a pool sample) with the trained generator. Then, they use gradient descent to minimize the objective. The latent variable minimizing the objective results in a synthetic image close to the separating hyper-plane. They annotate the synthetic sample and use it for training. Zhu & Bento (2017) demonstrate GAAL on subsets of MNIST and CIFAR-10 (two classes) using linear SVMs and DCGANs (Radford et al. (2015); Goodfellow et al. (2014)). However, GAAL performs worse than random sampling on both data sets, because it suffers from sampling bias and annotating is arbitrarily hard caused by sometimes poor quality of the synthetic uncertain samples. Note, that the GAAL requires visually distinct classes (horse & automobile) to enable manual annotations at all.

We propose ASAL that reuses the sample generation idea of Zhu & Bento (2017) but we use information entropy as uncertainty score and directly extend it to multiple classes. Additionally, ASAL uses CNN based classifiers instead of linear SVMs. For the generator we train Wasserstein GANs beforehand (Arjovsky et al. (2017)). We avoid annotating synthetic images by selecting the most similar ones from the pool with a newly developed sample matching method. We propose three different feature maps that we compute for each pool sample to fit a fast nearest neighbour model beforehand. During active learning, we compute the feature map of the synthetic sample and retrieve the most similar one from the pool in sub-linear time.

## 3 BACKGROUND

In this section we introduce the two uncertainty query strategies: minimum distance and maximum entropy sampling. We use the following notation: The set describing the pool is denoted by $\mathcal{P}$, the classifier at each AL cycle $k$ is denoted by $\theta^k$.

### 3.1 UNCERTAINTY SAMPLING

For uncertainty sampling where the model consists of a SVM the query strategy is based on the assumption that the model is least certain for samples that are in the vicinity of the separating hyper-

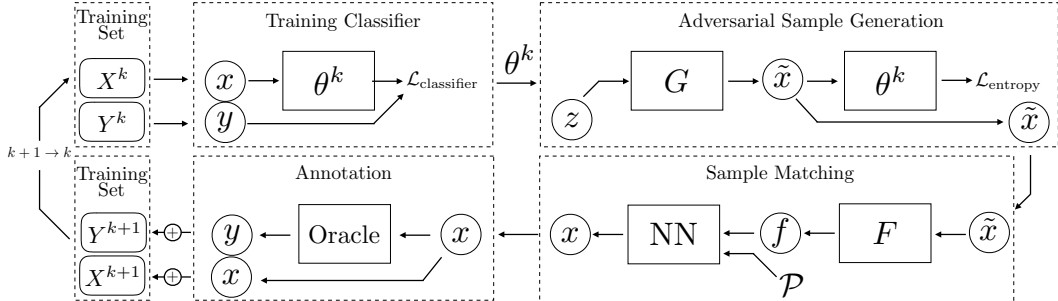

Figure 1: Block diagram of ASAL, where $(X^k, Y^k)$ with $(x, y)$ is the training set at cycle $k$, $\theta$ is the classifier, $z$ the latent variable, $G$ the generator, $\tilde{x}$ the synthetic samples, $F$ the feature extractor, $f$ the features, $\mathcal{P}$ the pool and NN the nearest neighbour method.

plane. Thus, newly selected samples are close to the decision boundary, are ideally support vectors that improve the decision boundary. Minimal distance sampling using SVM reads as

$$\begin{aligned} \text{minimize} \quad & \|(\mathbf{w}^k)^\top \phi(x) + b^k\|_2 \\ \text{subject to} \quad & x \in \mathcal{P}. \end{aligned} \tag{1}$$

where $\mathbf{w}$ and $b$ define the separating hyper-plane and $\phi(\cdot)$ is a feature map, e.g. induced by a SVM kernel or a neural network. Instead of considering the distance to the separating hyper-plane, information entropy computes the information content in each sample for the current classifier. Thus, the classifier is uncertain for samples with a high entropy and these samples have a high information content for the task of improving the classifier. Maximum entropy sampling read as follows:

$$\begin{aligned} \text{maximize} \quad & H_{\theta^k}(x) \\ \text{subject to} \quad & x \in \mathcal{P}, \end{aligned} \tag{2}$$

where $H_{\theta^k}(x) := \sum_{i=1}^m P(c = i|x; \theta^k) \log[P(c = i|x; \theta^k)]$ and $m$ is the number of categories.

Solving the optimization problems requires an exhaustive search over the whole pool $\mathcal{P}$ that requires computing the uncertainty score for each sample. Furthermore, we need to recompute the uncertainty score in each AL cycle because updating the classifier invalidates the previous score. Thus, classical uncertainty sampling has a linear run time complexity $\mathcal{O}(|\mathcal{P}|)$ with respect to the pool size $|\mathcal{P}|$.

## 4 PROPOSED ADVERSARIAL SAMPLING FOR ACTIVE LEARNING

ASAL adapts the sample generation idea of Zhu & Bento (2017) to pool based active learning using multiple classes and information entropy to measure uncertainty. Fig. 1 shows the main components of the proposed ASAL. We use a labelled data set $(X^k, Y^k)$ to train the classifier $\theta^k$. Then, we use the trained classifier $\theta^k$ and the generator $G$ to produce uncertain samples $\tilde{x}$. The feature extractor $F$ computes features that the nearest neighbour model uses to retrieve the most similar real samples from the pool. Finally, an oracle annotates the new samples and adds them to the training set. Then, a new AL cycle starts.

In the remainder of this section, we introduce the adversarial sample generation and the sample matching method.

### 4.1 ADVERSARIAL SAMPLE GENERATION USING GANS

Instead of selecting uncertain samples from the pool, we follow Zhu & Bento (2017) and generate such samples using GANs, that we train on the pool beforehand. GANs enable to approximate the underlying data distribution of the pool where the discriminator $D$ ensures that the samples

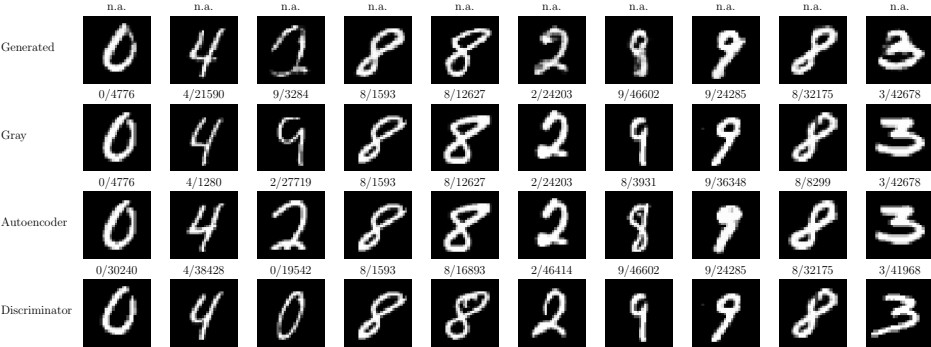

Figure 2: The rows show either generated or matched samples using different feature sets for *MNIST - ten classes*. The brackets denote (label id / sample id).

drawn from the generator $G$ are indistinguishable from real samples. At convergence, the generator produces the function $G : \mathcal{R}^n \to \mathcal{X}$ that maps the latent space variable $z \sim \mathcal{N}(\mathbf{0}_n, \mathbf{I}_n)$ to the image domain $\mathcal{X}$. Including the generator $G(\cdot)$ in Eq. equation 2 leads to the following optimization problem with respect to $x$

$$
\begin{aligned}
&\text{minimize} && \left(-H_{\theta^k} \circ G\right)(z) \\
&\text{subject to} && x = G(z).
\end{aligned}
\tag{3}
$$

Removing the constraint $x \in \mathcal{P}$ by including the generator simplifies the problem but changes its solution. New samples are no longer selected from the pool but are visually indistinguishable from these samples. We solve the optimization problem in two steps: (i) we use the chain rule and gradient descent to minimize the objective with respect to $z$ and (ii) we use $G$ to recover a synthetic sample $x$ from $z$. Thus, solving problem equation 3 has a constant run-time complexity $\mathcal{O}(1)$ because it is independent of the pool size.

## 4.2 SAMPLE MATCHING

The goal of the sample matching method is retrieving the most similar sample from the pool for a given synthetic sample. Thus, we need (i) representative features for comparison, (ii) a distance measure and (iii) a fast nearest neighbour method.

The ideal features would group the samples with similar entropy in features space close together. This guarantees that the nearest real neighbour of a synthetic sample with high entropy has a high entropy as well. However, updating the model, changes the entropy of each sample in the pool and destroys the previous structure in feature space. Thus, keeping a similar grouping in feature space, requires updating the features and recomputing the position of each sample. This leads to a linear run-time complexity. Hence, for a more efficient method we require fixed features for sample matching. To design such features, we use the fact that they are not required to structure the samples according to their entropy. Indeed it is sufficient that the features identify one sample in the pool that is very similar to the synthetic sample. Then, the two samples will not only share properties the classifier is comfortable with, but also the features that lead to high entropy. Thus, the features should be representative for the data set, be diverse and allow to discriminate the main properties of different samples.

The raw pixel values are a simple representation that allows to differentiate between different samples but is close for images with similar scene. Auto-encoders extract more representative features for a specific data set than the raw pixel values and lead to a compressed set of core features representing the images. Additionally, we study the features extracted from the discriminator that was used in the training of the GAN. We expect that the features used to differentiate between real and synthetic samples allow to compute representative sample properties.

We use the Euclidean distance measure to compute the similarity between two samples in feature space. Furthermore, we use a multidimensional binary search tree (k-d tree) (Bentley (1975)) for efficient nearest neighbour selection. The run-time complexity to search a nearest neighbour is sub-linear $\mathcal{O}(\log(|\mathcal{P}|)$ with respect to the pool size $|\mathcal{P}|$. Additionally, we use Principal Component Analysis (PCA) to reduce the number of dimensions of the feature space to achieve a small absolute run-time and to ensure similar run-times when using different features set with different number of dimensions.

## 5 EXPERIMENTS

### 5.1 DATASETS

For the experiments we use two different dataset: MNIST (LeCun et al. (1998)) and CIFAR-10 (Krizhevsky (2009)). The MNIST data set contains ten different digits 0 to 9 unevenly distributed. Each image has a resolution of $28 \times 28$ gray-scale pixels. The data set consists of 50k training, 10k validation and 10k testing samples. The CIFAR-10 consists of 50k training and 10k validation $32 \times 32$ color images with uniformly distributed label categories. We use the validation set for testing. For close comparison we follow Zhu & Bento (2017) and construct two class data sets, consisting of the MNIST digits 5 & 7 and the CIFAR-10 classes automobile & horse.

### 5.2 EXPERIMENTAL SETTINGS

First, we produce different references to assess the performance of ASAL. The classification accuracy for the *fully supervised* model establishes a performance upper bound that any active learning strategy attempt to approximate with as few training samples as possible. Furthermore, *random* sampling establishes the baseline that we want to exceed or at least perform equally. Additionally, we report the performance of traditional pool-based maximum entropy sampling that ASAL tries to approximate with sub-linear run-time complexity.

We examine three different versions of ASAL using the previously introduced set of features: *ASAL-Gray/RGB*, *ASAL-Autoencoder*, and *ASAL-Discriminator*. We reduce the dimension of the feature space to 50 using PCA. We experimentally verified that larger dimensions only increase the run-time but do not lead to better classification accuracy. To synthesize new samples we use the Adam optimizer and apply 100 gradient steps to minimize the negative entropy with respect to the latent space variable (see Eq. equation 3). Note, that we directly optimize for multiple latent space variables at the same time, embedding them in one batch with random initialization. We always draw samples from the pool without replacement. We do not use data augmentation for any experiment and train all models from scratch in each AL cycle. We run all experiments for five different runs with different random seeds except the computationally demanding experiments on CIFAR-10 with ten classes that we run for three random seeds. We report the training iterations for the GANs on each data sets in Tab. 3 in the appendix. We use the default values for all other parameters given by Gulrajani et al. (2017) and Wei et al. (2018a) in the papers and code (Gulrajani (2018); Wei et al. (2018b)). We describe the different architectures and training settings of the auto-encoders in Sec. B in the appendix. Additionally, we report further insights such as label distribution, entropy of newly added samples and additional experiments using other GANs or uncertainty scores in the appendix.

### 5.3 CLASSIFICATION RESULTS ON MNIST - TWO CLASSES

For binary digit classification we train a linear model with cross entropy loss. We train the model for 10 epochs using the Adam optimizer (Kingma & Ba (2015)) with a batch size of 10 and learning rate of 0.001. We train the Wasserstein GAN (Gulrajani et al. (2017)) with gradient penalty to synthesize only the digits 5 & 7. Fig. 3a shows that for a budget of 500 samples only the aggressive active learner reaches the performance of the fully supervised model. However, ASAL performs clearly superior to random sampling and converges faster to the accuracy of the fully supervised model. We want to emphasize, that all ASAL strategies outperform random sampling. Furthermore, Fig. 3b verifies that the entropy of newly added samples to the training set is higher for ASAL than for random sampling. On average, all versions of ASAL select samples with 63% higher entropy than randomly selected samples.

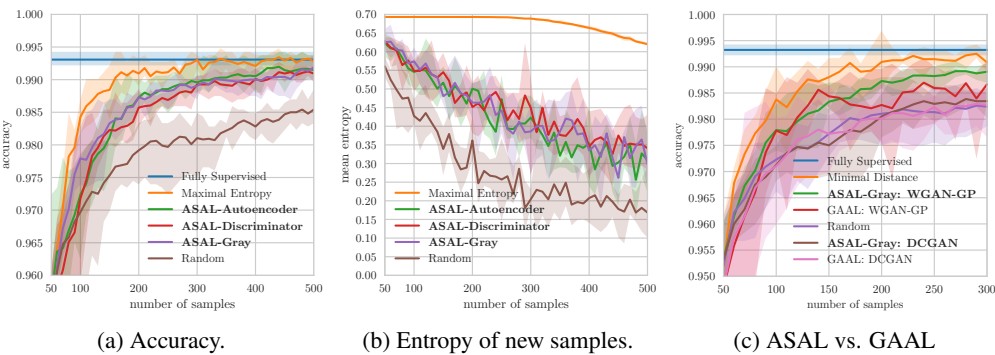

Figure 3: Test accuracy and entropy for different methods on *MNIST - two classes*.

### 5.3.1 PERFORMANCE COMPARISON BETWEEN THE PROPOSED ASAL AND GAAL

Zhu & Bento (2017) report worse performance than random sampling when training and testing on MNIST - two classes. For comparison we re-implement their GAAL with DCGAN. For a fairer comparison, we use an improved version of GAAL using the same Wasserstein GAN as ASAL and additionally test ASAL with the DCGAN. Fig. 3c shows that our implementation of GAAL performs similarly as random sampling and suffers less from sampling bias than reported by Zhu & Bento (2017). Possible reasons are the different human annotators or slightly different design choices. Furthermore, we observe that both methods outperform random sampling when using Wasserstein GANs and perform worse when using DCGAN. However, ASAL exceeds the performance of GAAL especially when using the Wasserstein GAN. Furthermore, using Wasserstein GAN leads to less stable performance of GAAL than ASAL(higher variance between 250 and 300 training samples for GAAL, note the red spikes in negative direction). Fig. 3c shows only the results for ASAL-Gray, for additional results, see Fig. 10 in the appendix.

## 5.4 CLASSIFICATION RESULTS ON CIFAR-10 - TWO CLASSES

For binary classification on CIFAR-10, we reuse the experimental setup presented in Sec. 5.3 but change the batch size to 50. We run the active learning strategies until the budget of 1000 samples is reached. Again we use a Wasserstein GAN (Gulrajani et al. (2017)) with gradient penalty that synthesizes only two classes (automobile & horse). Fig. 4b shows that especially ASAL-Autoencoder exceeds the performance of random sampling and achieves similar or slightly better results than exhaustive uncertainty sampling, for a training set containing more than 500 samples.

## 5.5 CLASSIFICATION RESULTS ON MNIST - TEN CLASSES

For ten digit classification we use LeNet (LeCun et al. (1998)) with cross entropy. We train the model for 10 epochs using the Adam optimizer with a batch size of 50 and learning rate of 0.001. For active learning, we start with an initial data set containing 10 samples for each class and add 50 samples each AL cycle. We run our experiments until the training set contains 10k samples. We synthesize samples for all ten classes using a Wasserstein GAN with gradient penalty Gulrajani et al. (2017). Fig. 4a shows that the proposed ASAL strategies also tackle multiple class problems and exceed the quality of random sampling.

## 5.6 CLASSIFICATION RESULTS ON CIFAR-10 - TEN CLASSES

Using all classes of CIFAR-10 complicates the classification task and we require a deep model to achieve close to state-of-the-art results. Therefore, we use the All-CNN model proposed by Springenberg et al. (2014) with a reported classification error of 9.08%. We use the proposed architectures and training strategies and use stochastic gradient descent with constant momentum of 0.9 and a learning rate of 0.01 that we decay by a factor of 10 at the 130th and the 140th epoch. We train the model for 150 epochs with a batch size of 128 without data augmentation and report a classification

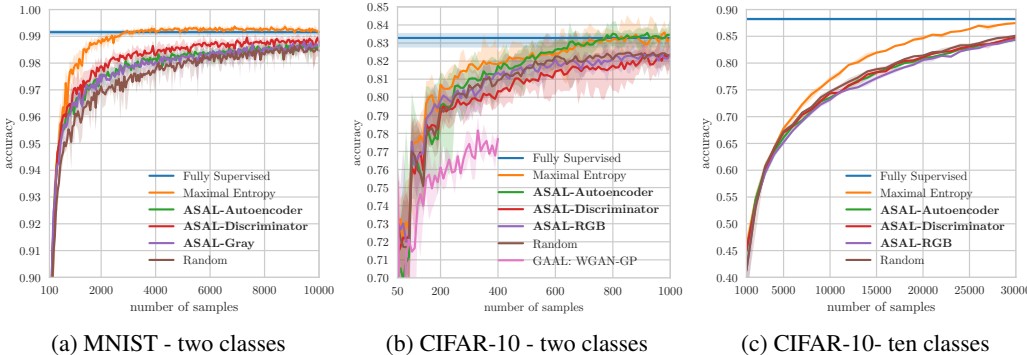

(a) MNIST - two classes   (b) CIFAR-10 - two classes   (c) CIFAR-10- ten classes

Figure 4: Test accuracy of ASAL on *MNIST* and *CIFAR-10*.

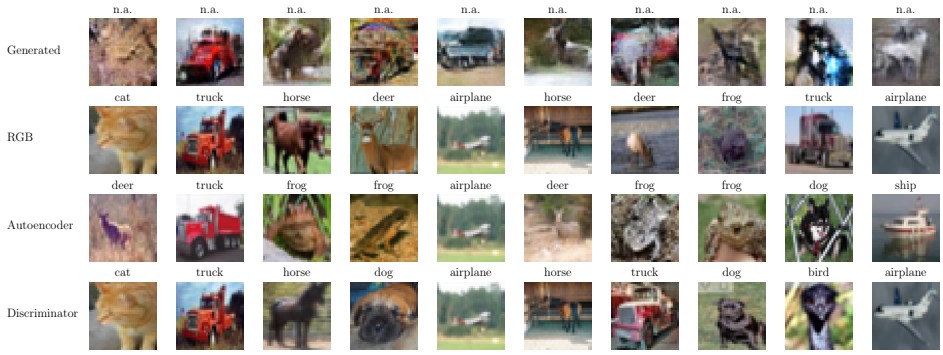

Figure 5: The rows show either generated or matched samples using different feature sets for *CIFAR-10 - ten classes*. The brackets denote (label id / sample id).

error of 11.8%. The All-CNN contains ∼1.4 million different parameters. Hence, we require larger initial training sets than for the previous models. Thus we include 100 randomly selected images per class. We add 1000 samples to the data set every AL cycle until the budget of 30k samples is reached. We generate ten times a batch containing 100 samples because optimizing for all samples at the same time is unfeasible. In contrast to the previous experiments we use a residual Wasserstein GAN (Wei et al. (2018a)) with gradient penalty and soft consistency term. We observed an Inception score of 7.8 without and 8.3 with adding the soft consistency term. We use the publicly available TensorFlow implementation of Wei et al. (2018b).

Fig. 4c shows the results for different ASALs using the promising residual GAN, that achieves the highest Inception score. Unfortunately, Fig. 4c shows that the performance of ASAL follows random sampling or is slightly worse, whereas maximum entropy sampling converges to the quality of the fully supervised model but uses 60% of all pool samples. Figs. 26 and 27 in the appendix show the label distribution for each AL cycle. It reveals that maximum entropy sampling selects most frequently cat exactly one of the classes that are least frequent in most of the training set of ASAL. Furthermore, Fig. 24 in the appendix reports the same experiments using different GANs but none of them leads to superior performance than random sampling.

## 5.7 DISCUSSION

Our experiments and results show that ASAL clearly outperforms random sampling and approximates exhaustive uncertainty sampling on three out of four benchmarks. Compared to GAAL, ASAL outperforms random sampling, enables annotating real samples, handles multiple class problems and uses CNN based classifiers. ASAL allows to update the feature maps of a classifier in each AL cycle and still achieves sub-linear run-time complexity whereas the hashing based methods

of Jain et al. (2010) has a linear run-time complexity if the feature maps are updated. Updating the classifier and keeping the features for matching fixed, leads to sub-linear run-times but without guaranteeing that newly added samples have the highest entropy of all samples available in the pool.

To achieve a sub-linear run-time complexity, ASAL requires to train a GAN and potentially an auto-encoder beforehand. Nonetheless, this initial cost pays off for extremely large data sets. Although, it might be impractical to consider each sample during training of the GAN, it can generate representative samples and ASAL allows to select samples from the pool that were not used to train the GAN. Thus, ASAL favours large data sets with similar samples, where it is only possible to train the GAN for a fixed number of iterations but contains a close match for any synthetic sample. Conversely, small data sets with diverse samples allow to train the GANs for many epochs such that it is align to the data distribution. However, real samples are sparsely distributed in feature space such that even the closest matches of a synthetic sample are significantly different.

We observed in Fig. 4c that ASAL performs similar as random sampling. Although ASAL enables to generate uncertain samples, it fails to select similar samples from the pool that have high entropy. One explanation is the aforementioned situation, where the images are diverse but the data set is comparatively small. Note, that CIFAR-10 is clearly more diverse than MNIST but has the same amount of samples. Furthermore, the top row in Fig. 5 shows that synthetic images still look unrealistic and identifying a similar real sample is a challenging problem. Another reason for poor performance is using low level features to compare different samples. To achieve state-of-the-art results on CIFAR-10, we had to use a much deeper network than for all other experiments but kept the architectures of the feature extractors almost identical. This can lead to a mismatch where the entropy of a sample mainly depends on high-level features but the matching method uses only low-level features to compare samples. Fig. 26 in the appendix shows for example that exhaustive uncertainty sampling selects most frequently images with the category cat exactly a class that ASAL selects least frequently. This is a sign that ASAL considers low-level features to find similar samples instead of more complex properties that characterize class information. Fig. 5 provides again such an indication. The last column shows a synthetic image with a white horse on a gray background and ASAL proposes matches with white object on a gray background but contain either a ship or an airplane. This means, that the classifier requires samples of a specific class it is uncertain about, ASAL generates these samples but fails to retrieve matches showing theses categories.

On *CIFAR-10 - two classes* we reported for ASAL-Autoencoder similar or slightly higher accuracy than for exhaustive uncertainty sampling. Although we consider uncertainty sampling as a performance reference that we try to approximate, it is always possible to exceed its performance. Note, entropy is one particular property that can identify informative samples. Nonetheless, it is possible that samples with lower entropy are more effective for training the classifier.

## 6 CONCLUSION

We proposed and evaluated a new pool-based active learning method that uses sample generation and matching. However, the sub-linear run-time complexity requires relaxing the guarantee, that selected samples have the highest entropy of all pool samples. We showed, that the success of ASAL depends on different factors: the structure of the data set, the quality of the trained GAN and the relevance of the feature used to compare samples. A poor GAN can generate high entropy samples but poor quality samples are impractical to match. Small data sets that contain very different samples complicate both, training GANs and finding similar matches. Less representative features might not contain the properties needed to find similar samples, where both have a high entropy. Nonetheless, we demonstrated that ASAL outperforms random sample selection and approximates exhaustive uncertainty sampling in three out of four cases. Furthermore, the sub-linear run-time complexity makes ASAL suitable for large data set. We pointed out that ASAL uses low-level feature but there are signs that high-level features might be more suitable to match samples. Thus, one particular direction of future research includes identifying such high-level features. Possible candidates are VGG (Simonyan & Zisserman (2015)) or AlexNet (Krizhevsky et al. (2012)) features. Training the model on the unlabelled pool and the small initial data set might lead to features covering the needed properties. In addition, sample generation allows adding other scores beside information entropy. Thus, an interesting direction of future research is designing other scores that will be used during sample generation i.e. measuring sample diversity (Zhu et al. (2003); Xu et al. (2007).

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

# A    ADDITIONAL EXPERIMENTS ON CELEBA

To further strengthen the experiments presented in the main paper we create three benchmarks using CelebA and report the performance of ASAL in terms of test accuracy and run-time.

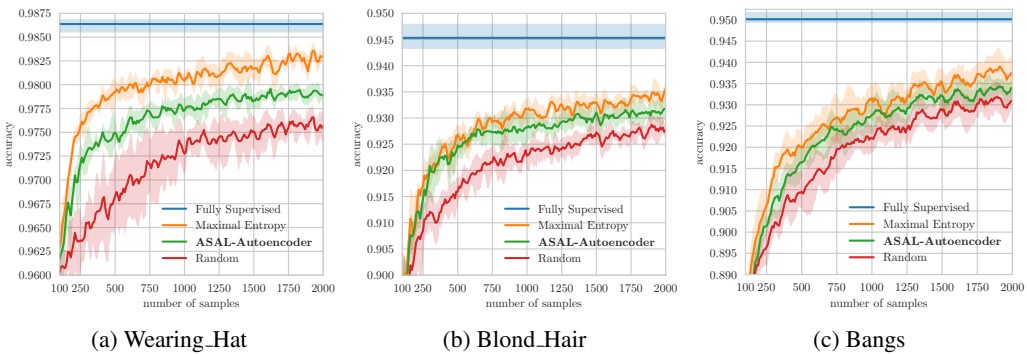

| (a) Wearing_Hat | (b) Blond_Hair | (c) Bangs |
|---|---|---|

Figure 6: Test accuracy of three different classification benchmarks constructed from CelebA. The data set is separated in images where the attribute is present or absent.

## A.1    DATA SET

CelebA consists of roughly 160k training, 20k validation and 20k testing images. Thus, it is more than $3\times$ larger than MNIST or CIFAR-10. We use the provided 40 face attributes to construct three different classification benchmarks. Each benchmark contains all 200k images labelled according to the presence or absence of the corresponding face attribute (Wearing Hat, Bangs and Blond Hair). We keep the suggested splitting into training, validation and testing for each benchmark.

## A.2    EXPERIMENTAL SETUP

We train a Wasserstein GAN with gradient penalty and an auto-encoder for ASAL beforehand (see Tabs. 1 and 2 for the architectures). We use a Nvidia GeForce GTX TITAN X GPU to train the models. The 100k training iterations for the GAN take roughly 25 h and the 50k iterations of the auto encoder 1.6 h. We use for both the Adam optimizer with a learning rate of 0.0001 and batch size 64. The number of compressed features of the auto-encoder is 128. For sample matching we decrease the number of features to 50 using PCA.

For classification we use the CNN presented in Tab. 1. We use the Adam optimizer with a learning rate of 0.001 and a batch size of 50 and train for 30 epochs. We start active learning with 100 labelled samples, where the number of samples per class corresponds to the data distribution. We select and label ten new samples in each active learning cycle until we exhaust the budget of 2000 samples. We run all experiments for three different random seeds. For sample generation we apply 100 gradient descent steps using the Adam optimizer with a step size of 0.01. We optimize for all ten samples at the same time.

## A.3    RESULTS AND TIMINGS

Fig. 6 shows the test accuracy of random sampling, maximum entropy sampling and ASAL for three different benchmarks on CelebA. We conclude, that ASAL clearly outperforms random sampling and approaches the accuracy of maximum entropy sampling but requires less time for sample selection, see Fig. 7. Thus, the proposed idea of sample generation and matching is simple but effective.

Fig. 7 reports the time required to select ten new samples in each active learning cycle with respect to different data set sizes. We randomly augmented the data set containing 160k samples to report the timings of maximal entropy sampling and ASAL (only nearest-neighbor search depends on data set size) for data sets with up to 16M samples. Whereas it it possible to keep all images in memory for 160k (1.98GB) this is hardly possible for 16M images (198GB). Nonetheless, we did

not add additional I/O-time for reading the images from disk to the maximum entropy timings in each AL cycle. The sample matching proposed in ASAL reduces the memory consumption because it requires to keep only 50 features per images (32MB for 160k and 3.2GB for 16M images). This saving allows to keep the features in memory and to build the nearest neighbor model even for huge data sets. We use a Nvidia GeForce GTX TITAN X GPU to report all timings.

ASAL has a sub-linear run-time complexity to select new samples. However, it requires several pre-processing steps such as training the GAN ($\sim 25$h), training the auto-encode ($\sim 1.6$h), extracting the features ($\sim 32$s per 160k samples) and fitting the nearest-neighbor model ($\sim 5$min for 16M samples). Note, that the number of iterations depends mainly on the difficulty of the data set than on its size. As sample selection for ASAL is almost negligible ( 44s for 16M samples per AL cycle) compared to the pre-processing time, the transition point is approximately then, when uncertainty sampling takes more time than preparing ASAL. Thus, maximal uncertainty sampling is more efficient when using small data sets or running active learning only for a few samples. Nonetheless, for all settings and data set sizes there exists a transition point, see Fig. 8. For example, ASAL is already more efficient after only 30 cycles (300 added samples) than maximum entropy sampling for a data set containing 16 million samples for the given setup.

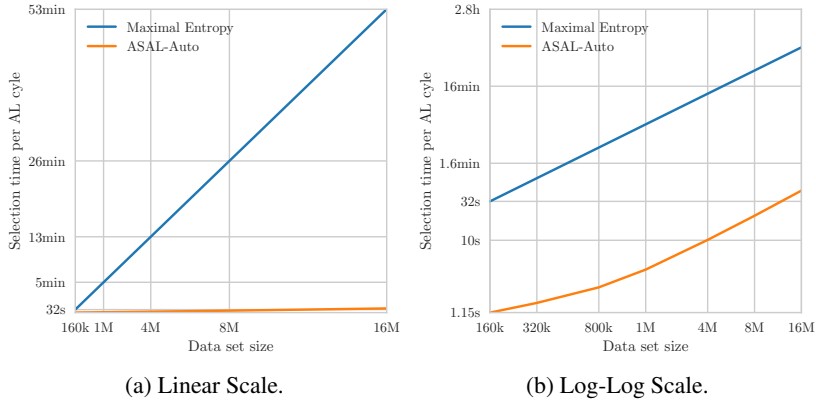

(a) Linear Scale.  (b) Log-Log Scale.

Figure 7: Run-time of maximal entropy sampling and ASAL to select 10 new samples in one AL cycle with respect to the data set size. The numbers are established using the setup described for CelebA running on a Nvidia GeForce GTX TITAN X GPU.

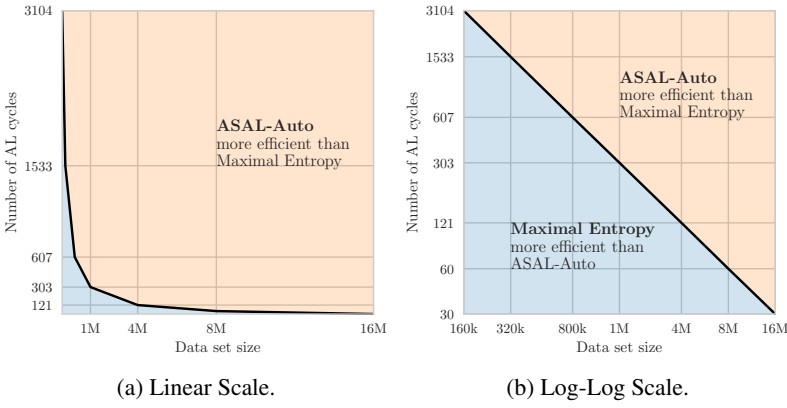

(a) Linear Scale.  (b) Log-Log Scale.

Figure 8: ASAL is more efficient for selecting new samples than maximal entropy sampling. However, it requires pre-processing time. These diagrams show the transition point (number of AL cycles when ASAL gets more efficient than maximum entropy sampling) with respect to the data set size.

Table 1: Model architectures for ASAL on CelebA.

| Classifier | Generator | Discriminator |
|---|---|---|
| Input: $64 \times 64 \times 3$ | Input: $z \sim \mathcal{N}(0, 1)$: 128 | Input: $64 \times 64 \times 3$ |
| $3 \times 3$ conv: 16 
 ReLU, Maxpool $2 \times 2$ | linear: $128 \times 4096$ 
 Batch norm, ReLU | $5 \times 5$ conv: 128, stride=2 
 leakyReLU |
| $3 \times 3$ conv: 32 
 ReLU, Maxpool $2 \times 2$ | $5 \times 5$ deconv: 256 
 Batch norm, ReLU | $5 \times 5$ conv: 256, stride=2 
 leakyReLU |
| $3 \times 3$ conv: 64 
 ReLU, Maxpool $2 \times 2$ | $5 \times 5$ deconv: 128 
 Batch norm, ReLU | $5 \times 5$ conv: 512, stride=2 
 leakyReLU |
| linear: $4096 \times 1024$ 
 ReLU, Dropout 0.5 | $5 \times 5$ deconv: 64 
 Batch norm, ReLU | $5 \times 5$ conv: 512, stride=2 
 leakyReLU |
| linear: $1024 \times 1$ | $5 \times 5$ deconv: 3 
 Tanh | linear: $8192 \times 1$ |

Table 2: Auto-encoder architecture for ASAL on CelebA.

| Encoder | Decoder |
|---|---|
| Input: $64 \times 64 \times 3$ | Input: $4 \times 4 \times 16$ |
| $5 \times 5$ conv: 128, stride=2 
 Batch norm, ReLU | $5 \times 5$ deconv: 32 
 Batch norm, ReLU |
| $5 \times 5$ conv: 64, stride=2 
 Batch norm, ReLU | $5 \times 5$ deconv: 64 
 Batch norm, ReLU |
| $5 \times 5$ conv: 32, stride=2 
 Batch Norm, ReLU | $5 \times 5$ deconv: 128 
 Batch norm, ReLU |
| $5 \times 5$ conv: 16, stride=2 | $5 \times 5$ deconv: 3 
 Tanh |

# B ARCHITECTURES AND TRAINING OF AUTO-ENCODERS FOR ASAL

For MNIST - two classes we use the following auto-encoder settings: the encoder consists of three convolution layers with a stride of two, each followed by an activation leading to 64 compressed features. The decoder uses three deconvolution layers each with an activation. We train the auto-encoder for 40 epochs with a batch size of 100 using the Adam optimizer with a learning rate of 0.001. For MNIST - ten classes we reuse the same settings but with three times more channels resulting in 192 compressed features.

The Encoder for CIFAR-10 consists of three layers, each with a convolution followed by batch normalization (Ioffe & Szegedy (2015)), activation and max pooling (stride of two and window size $2 \times 2$). The number of compressed features is 256. The decoder uses first a layer consisting of a convolution, batch normalization and activation followed by three deconvolution layers each with batch normalization and activation. We train the auto-encoder for 100 epochs with a batch size of 128 using the Adam optimizer with a learning rate of 0.001. We use the same settings for CIFAR-10 - two classes and CIFAR-10 - ten classes.

# C ADDITIONAL RESULTS: MNIST - TWO CLASSES

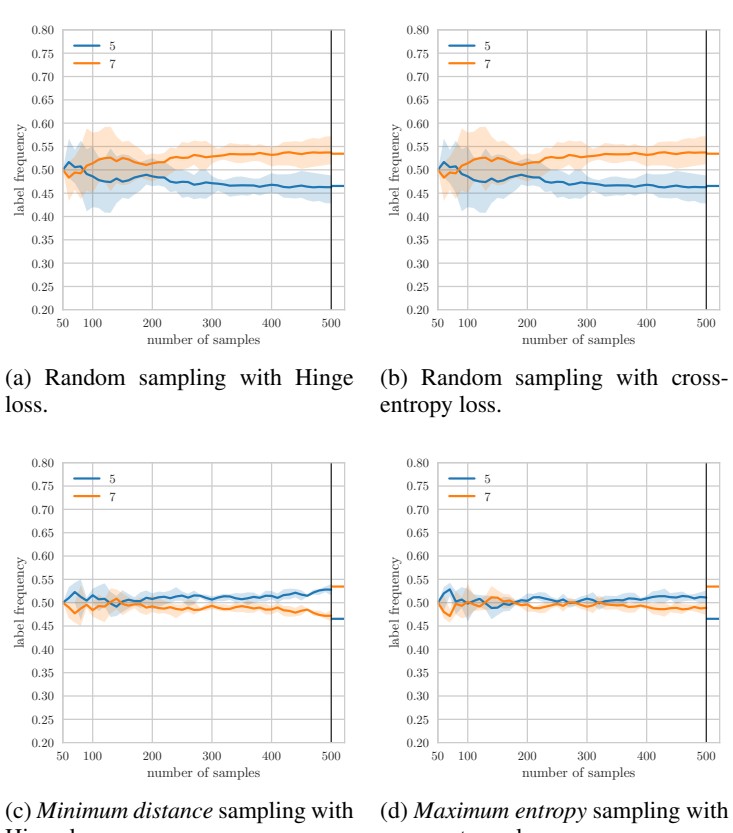

(a) Random sampling with Hinge loss.

(b) Random sampling with cross-entropy loss.

(c) *Minimum distance* sampling with Hinge loss.

(d) *Maximum entropy* sampling with cross-entropy loss.

Figure 9: Label distribution for uncertainty sampling using maximum entropy and random sampling for *MNIST - two classes* using different uncertainty measures and loss functions. The tick on the right show the true label distribution in the pool. The label distribution of the training set, assembled with random sampling (top), converges to the true label distribution of the pool. Conversely, uncertainty sampling leads to a training set that contains more frequently the label 5 than 7 compared to the pool that contains 7 more frequently. Apparently, images with the digit 5 lead to higher uncertainty of the used classifier.

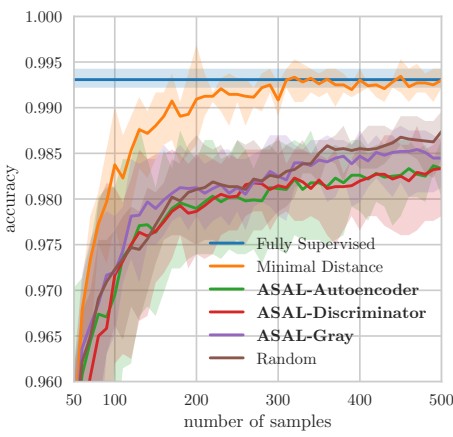

(a) *Minimum distance* with Hinge loss and DC-GAN.

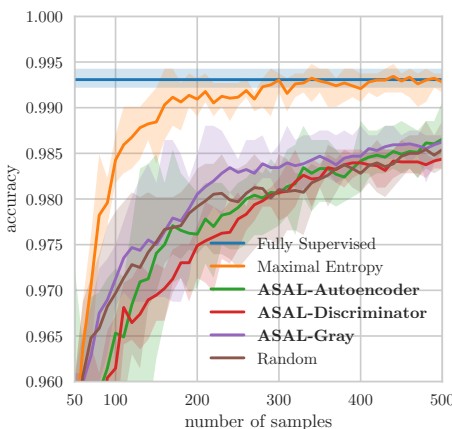

(b) *Maximum entropy* with cross-entropy loss and DCGAN.

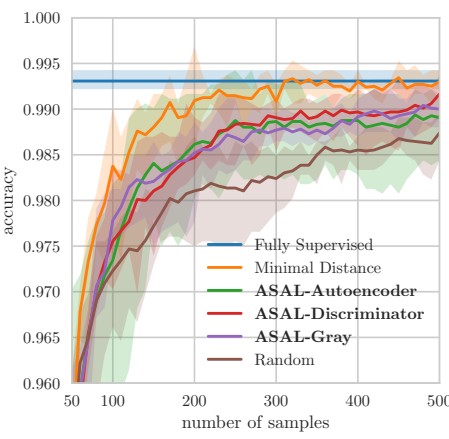

(c) *Minimum distance* with Hinge loss and WGAN-GP.

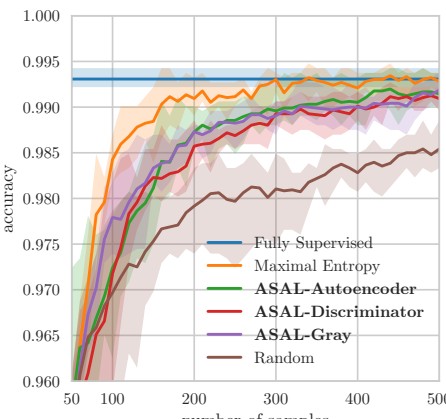

(d) *Maximum entropy* with cross-entropy loss and WGAN-GP.

Figure 10: Test accuracy on *MNIST - two classes* of a fully supervised model, for random sampling, uncertainty sampling and different ASALs using different GANs, uncertainty measures and loss functions. ASAL with WGAN-GP (bottom) clearly exceed the performance of ASAL using DCGAN (top). Maximum entropy sampling and using the cross entropy loss lead to the setup (10d) that approaches the fully-supervised model with the fewest samples and reaches the smallest gap for all ASAL using 500 labelled samples.

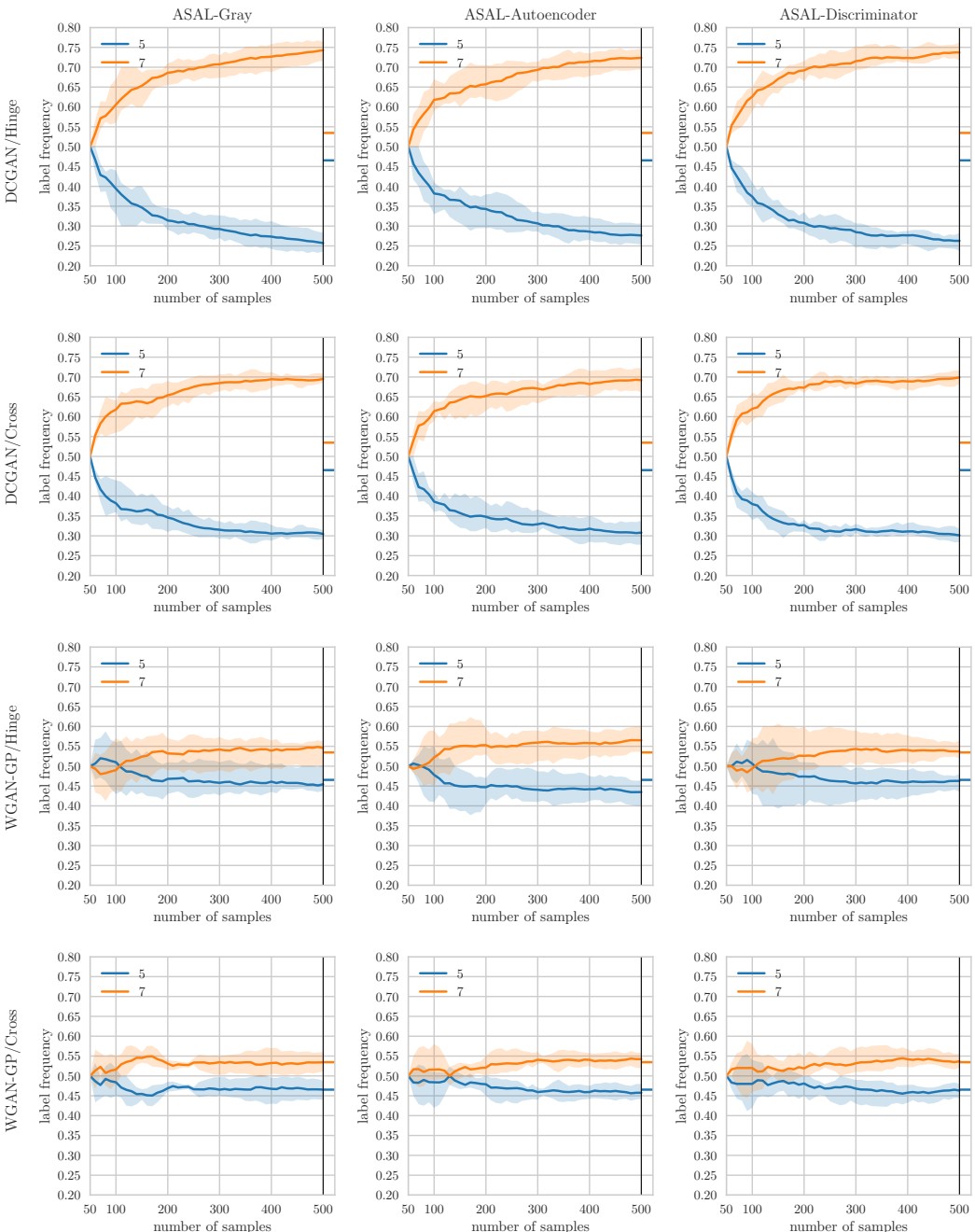

Figure 11: Label distribution for active learning using different matching strategies, uncertainty measures and GANs for *MNIST - two classes*. The ticks on the right show the true label distribution in the pool. ASAL using WGAN-GP (third and fourth row) reaches a label distribution of the training data that is similar to the true label distribution in the pool. Conversely, ASAL using DCGAN (first and second row) leads to a training set that contains almost three times as many images with the digit 7 than digit 5. Most likely, the DCGAN is responsible for this behaviour because we already observed that it produces the digit 7 more frequently than the digit 5, see Fig. 32a.

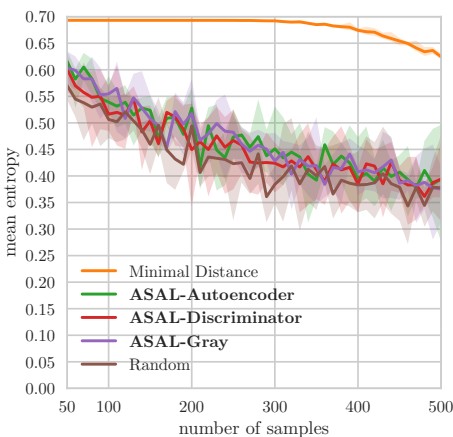

(a) *Minimum distance* with Hinge loss and DC-GAN.

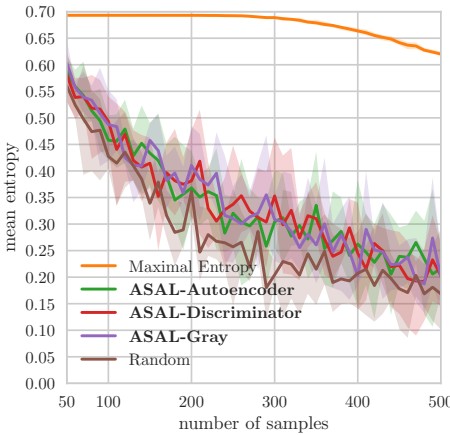

(b) *Maximum entropy* with cross-entropy loss and DCGAN.

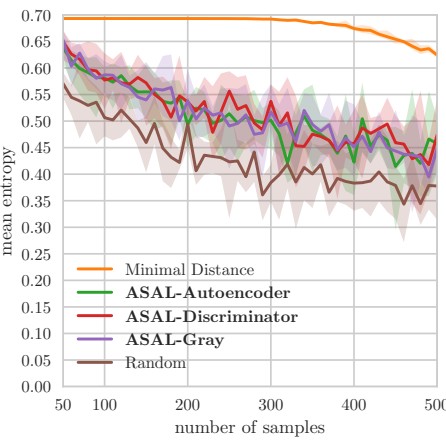

(c) *Minimum distance* with Hinge loss and WGAN-GP.

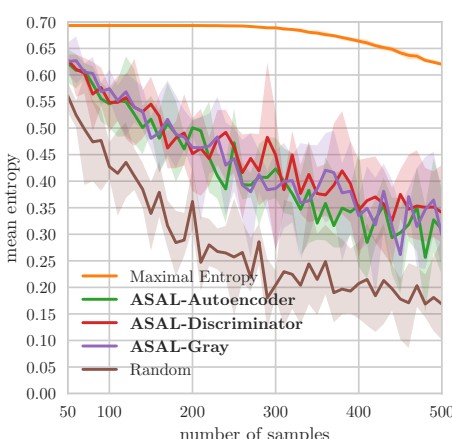

(d) *Maximum entropy* with cross-entropy loss and WGAN-GP.

Figure 12: Average entropy of images that are selected and added to the training set for *MNIST - two classes* using different GANs, uncertainty measures and loss functions. All figures show that ASAL selects samples from the pool that have a higher entropy than randomly sampled images. However, maximum entropy sampling and WGAN-GP (12d) lead to the largest entropy gap between selected and randomly sampled images. Maximum entropy sampling (right column) results in smaller average entropy of the classifier than minimum distance sampling (left column) because we use the cross-entropy loss that directly optimizes for small entropy, opposed to the hinge loss that minimizes the distance to the separating hyper-plane.

## C.1 Agreement of Manual Annotations and Matched Labels

Instead of manually annotating images we propose to select similar images from the pool and ask for labels of these images. Similar images might show an object of the same class, have similar surrounding, colors, size or share other features. Thus, we compare the agreement of the manual class annotations of the generated images with the matched images, using the three different strategies. We use 1300 generated samples for each GAN, annotate the images manually and retrieve the closest match with the corresponding label from the pool. We assume that the final model will be measured on an almost evenly distributed test set similar to MNIST and USPS. However, the test set for this experiment contains the generated samples with manual annotations and the GAN may generate samples with unevenly distributed label frequency. Thus, we compute the accuracy for each class independently and average these values subsequently to obtain the final score.

Fig. 13 shows that the agreement is higher for ASAL strategies using WGAN-GP than DCGAN. Furthermore, we observe that the matching based on gray values achieves the highest agreement. Similarly, Figs. 10a and 10b show best performance for ASAL-Gray.

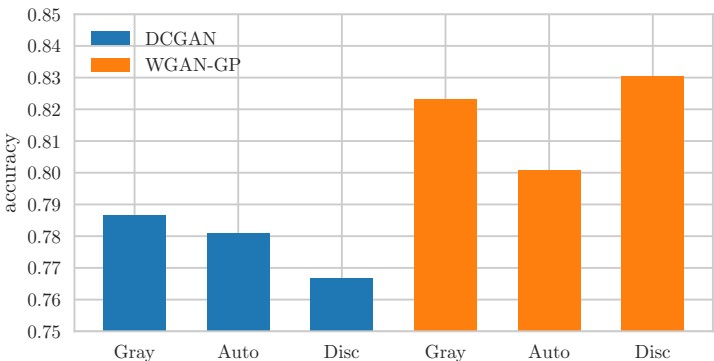

Figure 13: Comparison of the agreement accuracy between manual annotations and matched labels. The matching strategies employed in ASAL allow to select similar images from the pool and compare these labels to the manual annotations. For *MNIST - two classes* the agreement for WGAN-GP is higher than for DCGAN.

Table 3: Number of training iterations for the different GANs and data sets.

| GAN | MNIST | | CIFAR-10 | |
|---|---|---|---|---|
| | two | ten | two | ten |
| DCGAN | 40k | 100k | 100k[2] | 200k |
| WGAN-GP | 40k | 100k | 200k | 200k |
| Residual WGAN-GP | — | — | 100k | 100k |
| Residual WGAN-CT | — | — | 100k | 100k |

---

[2]We observe mode collapse for *CIFAR-10 - two classes* using DCGAN when training the GAN for more than 100k iterations.

# D    ADDITIONAL RESULTS: MNIST - TEN CLASSES

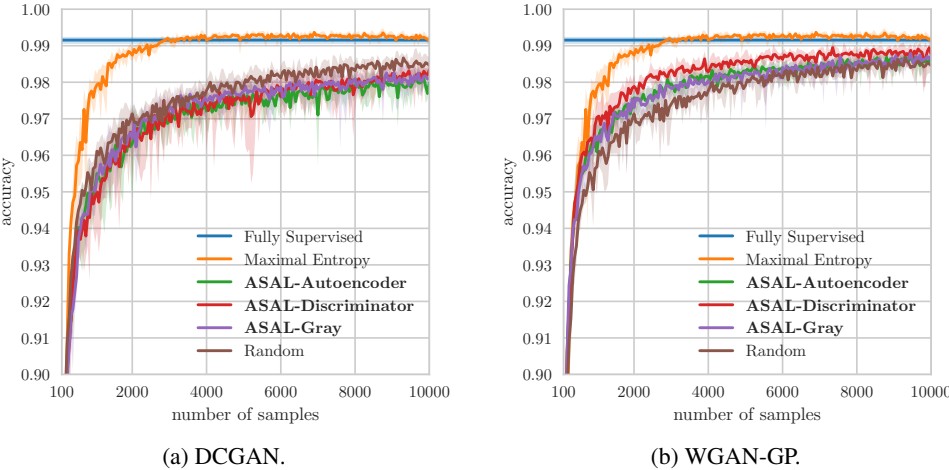

(a) DCGAN.

(b) WGAN-GP.

Figure 14: Test accuracy on *MNIST - ten classes* of a fully supervised model, for random sampling, uncertainty sampling and different ASALs using two different GANs. Selecting new images using random samples exceeds the performance of the proposed strategy when using the DCGAN. However, replacing the DCGAN with the WGAN-GP enables outperforming random sampling. ASAL-Discriminator achieves the best quality.

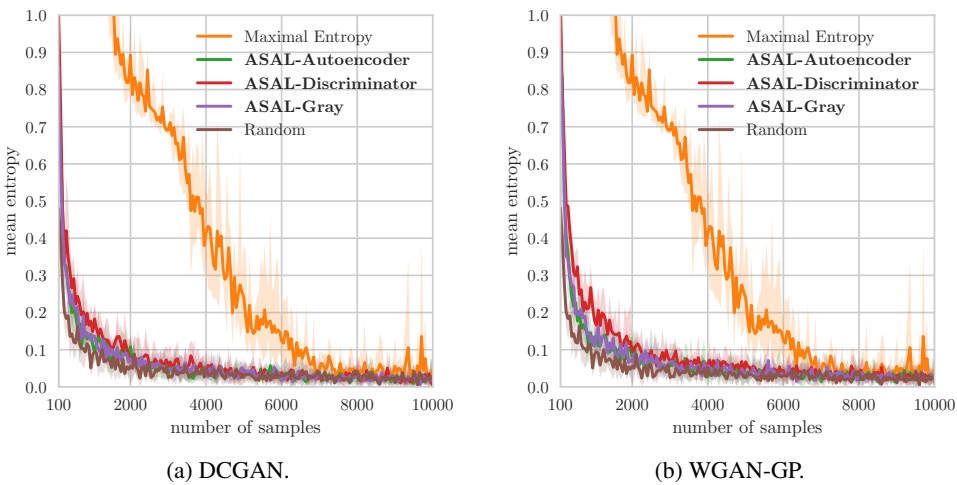

(a) DCGAN.

(b) WGAN-GP.

Figure 15: Average entropy of images that are selected and added to the training set for *MNIST - ten classes* using different GANs. Both figures show that at the beginning ASAL selects images with higher entropy than random sampling. In average WGAN-GP leads to a larger gap than DCGAN. However, this gap rapidly shrinks when increasing the training set.

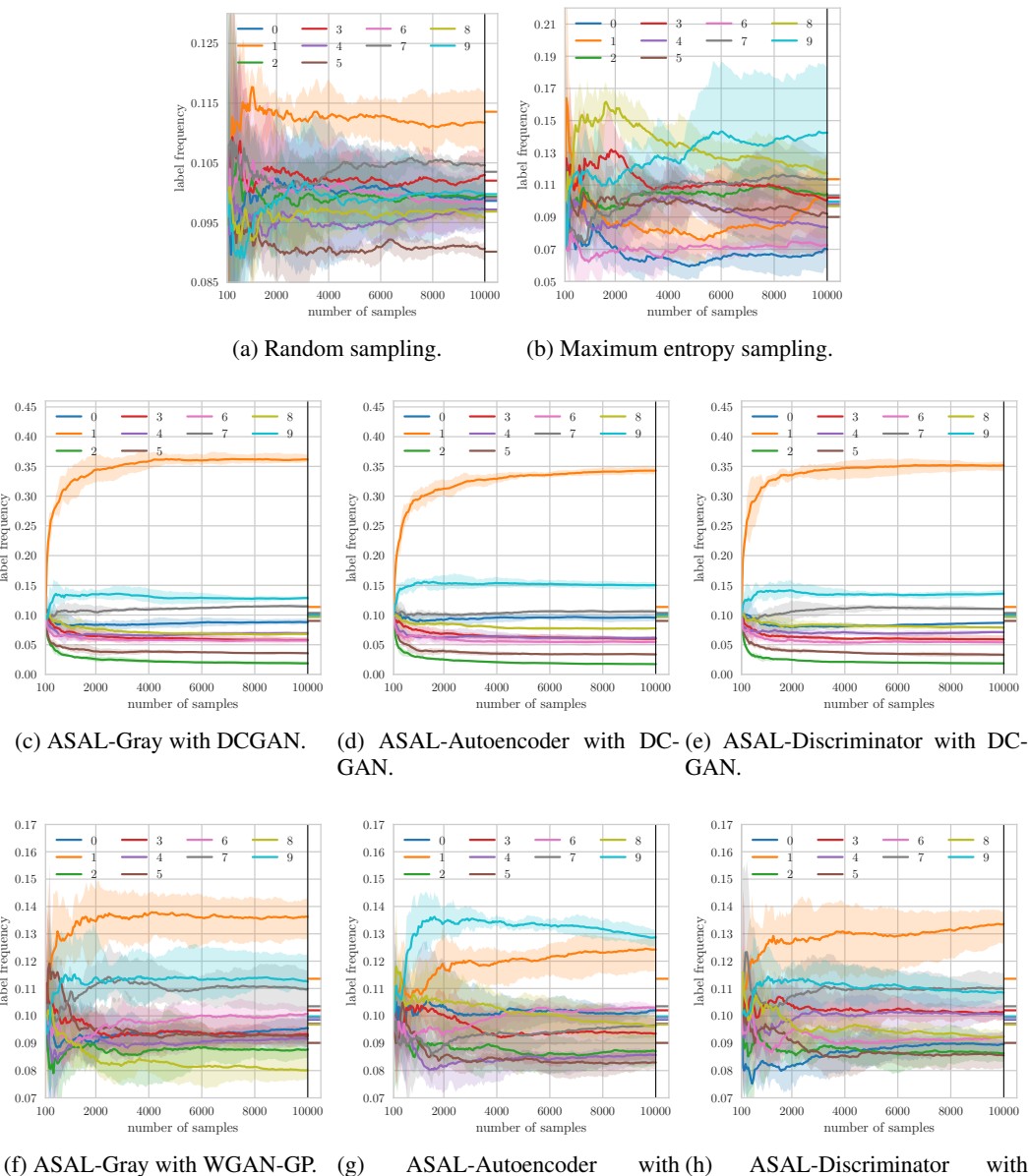

Figure 16: Label distribution for uncertainty sampling using maximum entropy, random sampling and active learning using different matching strategies and GANs for *MNIST - ten classes*. The tick on the right show the true label distribution in the pool. Note the different scaling of the y-axis. Random sampling converges to the true label distribution in the pool and maximum entropy sampling leads to a training set with a higher ration of certain digits (7,8,9) or lower (0,1,4,6) than the pool. Similarly, ASAL using WGAN-GP (bottom row) selects certain digits more frequently than others. Conversely, ASAL using DCGAN (top row) leads to a training set that contains 30% images with the digit 1. Most likely, the DCGAN is responsible for this behaviour because we already observed that it produces the digit 1 more frequently than any other digit, see Fig. 33a.

# E   TRAINING ON MNIST AND TESTING ON USPS

Zhu et al. Zhu & Bento (2017) report the accuracy of GAAL when trained on MNIST and tested on USPS for two classes. They report best performance on USPS and outperform the fully supervised model. However, it is unclear how they up-sample the $16 \times 16$ USPS images to test on the $28 \times 28$ model trained on MNIST. We redo the experiments using ASAL and up-sample the USPS images as follows: (1) padding the images with three pixels at each side, (2) up-sampling the images to 30 and (3) cropping the images to $28 \times 28$ to remove boundary artifacts. Following this strategy, we report an average test accuracy of 0.91 for the fully supervised model compared to 0.70 reported by Zhu et al. Zhu & Bento (2017). Fig. 17 shows that ASAL outperforms aggressive uncertainty and random sampling.

We repeat the experiment for MNIST - ten classes using DCGAN and WGAN-GP. This time, uncertainty sampling clearly outperforms all other strategies and the fully supervised model. Nonetheless, ASAL-Auto and ASAL-Disc lead to a better training performance than passive learning for WGAN-GP.

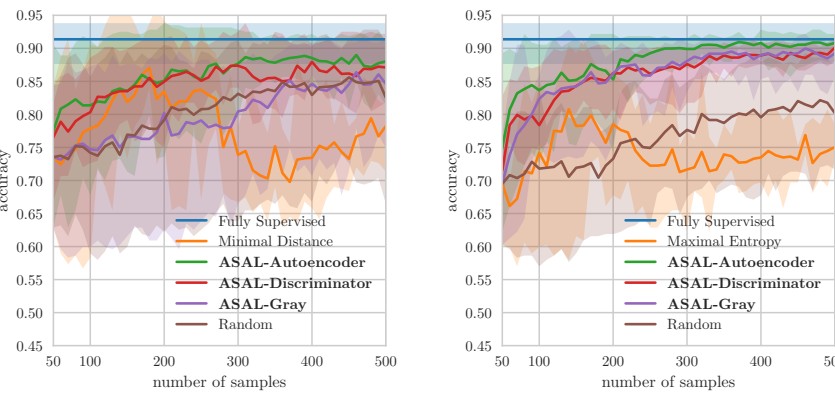

(a) *Minimum distance* with Hinge loss and DCGAN.

(b) *Maximum entropy* with cross-entropy loss and DCGAN.

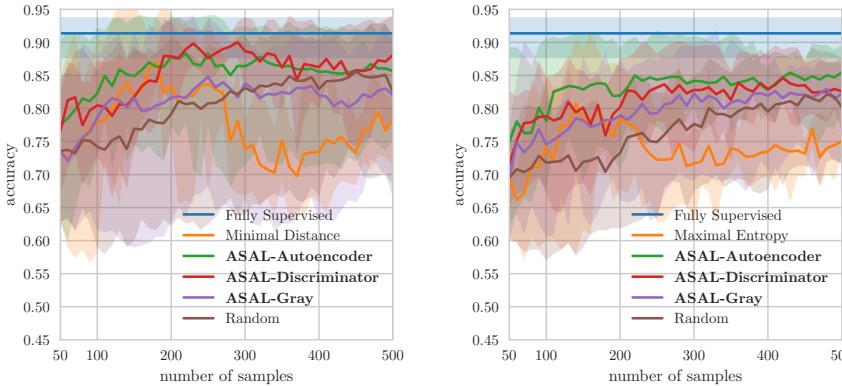

(c) *Minimum distance* with Hinge loss and WGAN-GP.

(d) *Maximum entropy* with cross-entropy loss and WGAN-GP.

Figure 17: Test accuracy on *USPS - two classes* but trained on *MNIST - two classes* of a fully supervised model, for random sampling, uncertainty sampling and different ASALs using different GANs, uncertainty measures and loss functions. Uncertainty sampling performs worse than any other strategy because it aggressively trains the classifier for the samples present in the pool and generalizes less. Random sampling and ASAL tend to generalize better by respecting the true data distribution either through random sampling or using a pretrained GAN on the data set to find new samples.

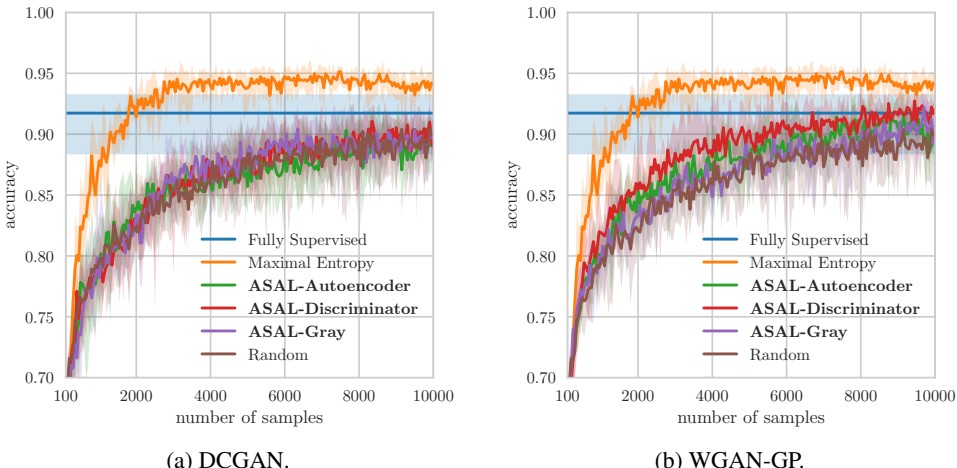

(a) DCGAN.                                    (b) WGAN-GP.

Figure 18: Test accuracy on *USPS - ten classes* but trained on *MNIST - ten classes* of a fully super-vised model, for random sampling, uncertainty sampling and different ASALs using two different GANs. Maximum entropy sampling for ten classes exceeds the quality of any other method compared to binary classification where it performed worst, see Fig. 17. The more elaborate LeNet and using more classes and samples to train lead to a classifier that generalizes well. The active learning strategies using WGAN-GP exceed the quality of random sampling. ASAL-Disc. even outperforms the fully supervised mode. ASAL using DCGAN performs comparable to random sampling.

## F    ADDITIONAL RESULTS: CIFAR - TWO CLASSES

For CIFAR-10, we do not indicate the true label distribution by a tick because the validation set contains the same number of samples for each class.

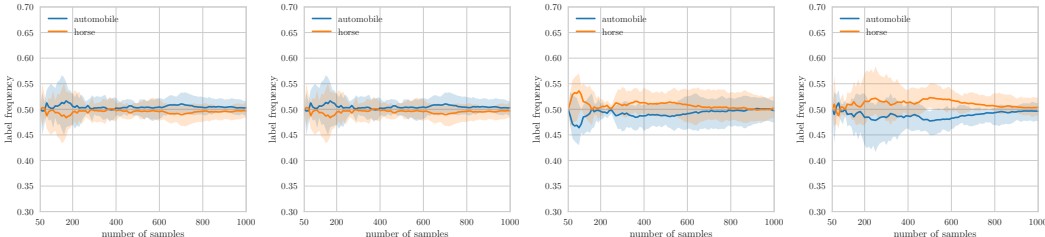

(a) Random sampling with Hinge loss.  (b) Random sampling with cross-entropy loss.  (c) *Minimum distance* sampling with Hinge loss.  (d) *Maximum entropy* sampling with cross-entropy loss.

Figure 19: Label distribution for uncertainty sampling using maximum entropy and random sampling for *CIFAR-10 - two classes* using different uncertainty measures and loss functions. The label distribution of the training set of all strategies converges to the true label distribution of the pool. However, in average over all active learning iterations the training set of the uncertainty sampling strategies most frequently contained the images with the label horse.

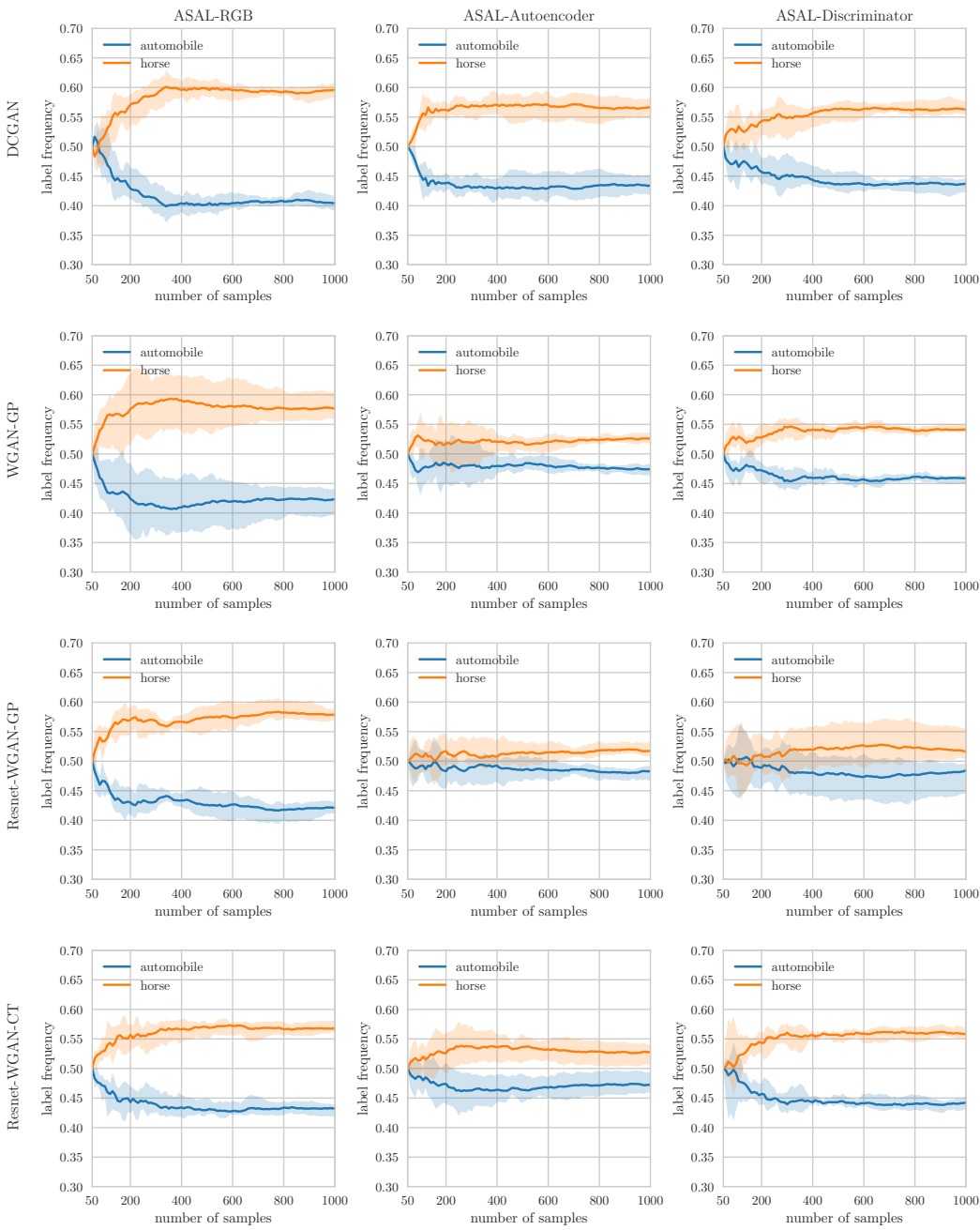

Figure 20: Label distribution for active learning with minimum distance sample generation and the Hinge loss, using different matching strategies and GANs for *CIFAR-10 - two classes*. All setups assemble training sets containing the more image with the label horse than automobile.

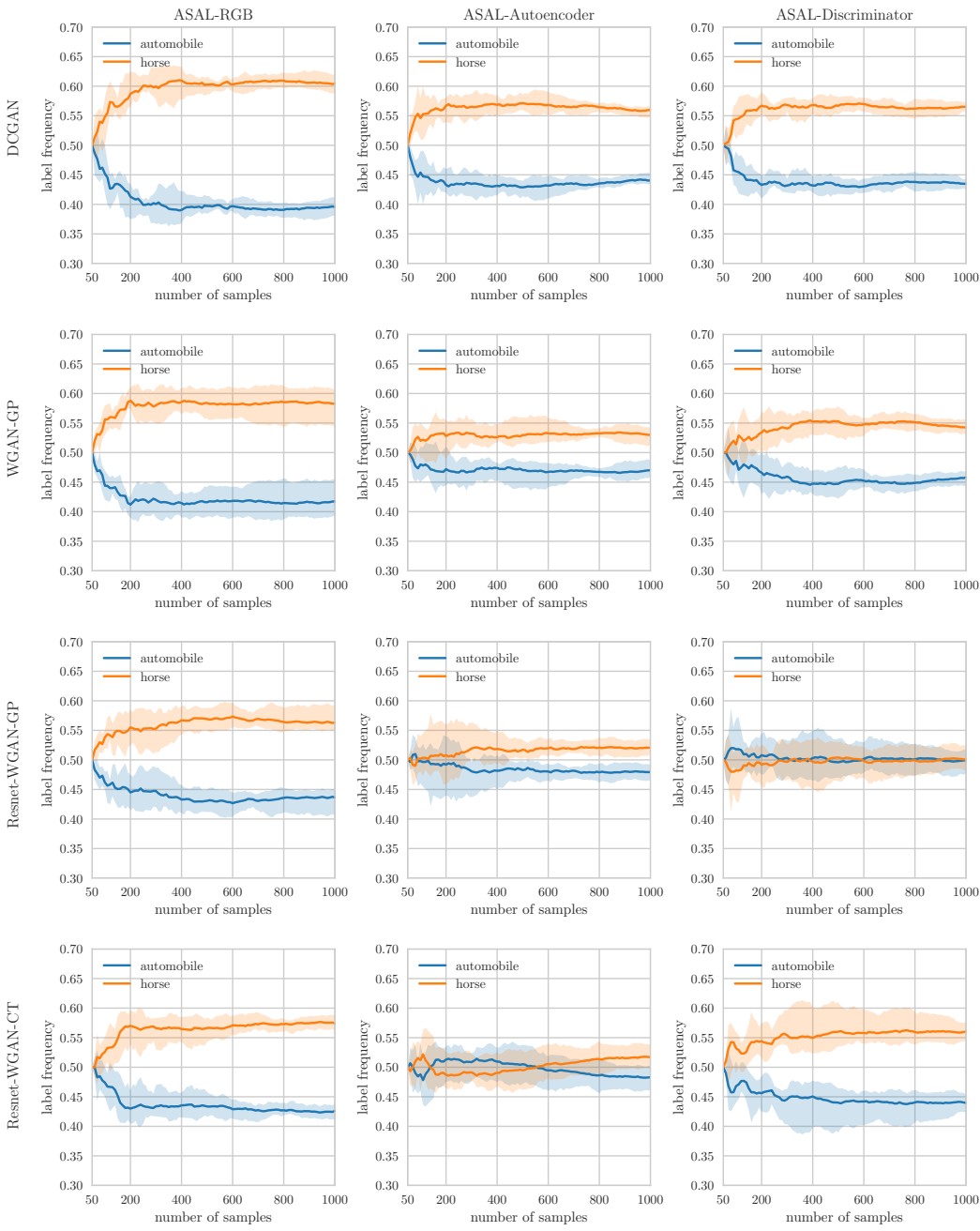

Figure 21: Label distribution for active learning with maximum entropy sample generation and the cross-entropy loss, using different matching strategies and GANs for *CIFAR-10 - two classes*.

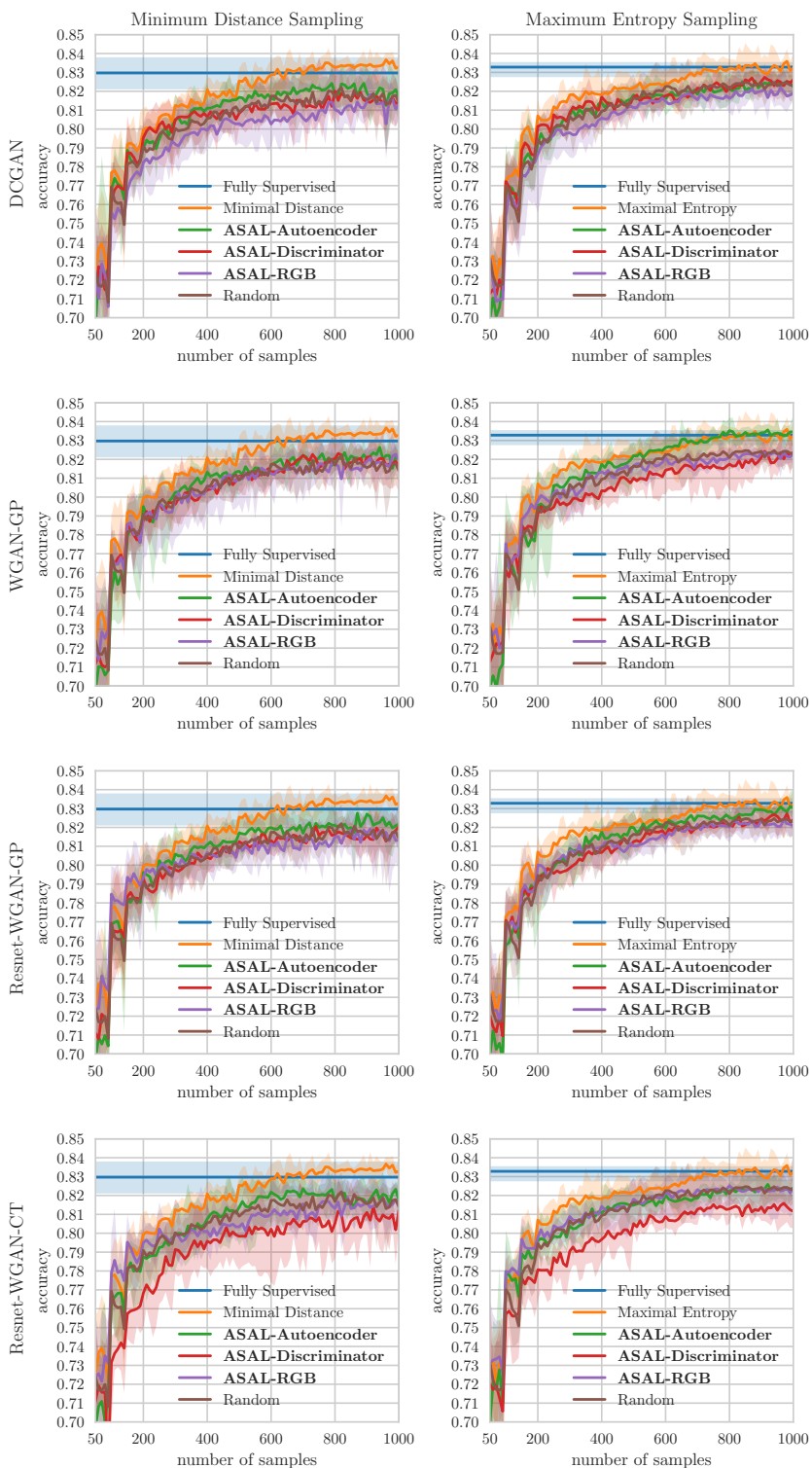

Figure 22: Validation accuracy on *CIFAR-10 - two classes* of a fully supervised model, for random sampling, uncertainty sampling and different ASALs using different GANs. ASAL-Autoencoder leads to the best performance. ASAL-Disc. using Resnet-WGAN-CT performs worse that any other strategy because the sample matching using is unable to retrieve high entropy samples from the pool, see Fig. 23.

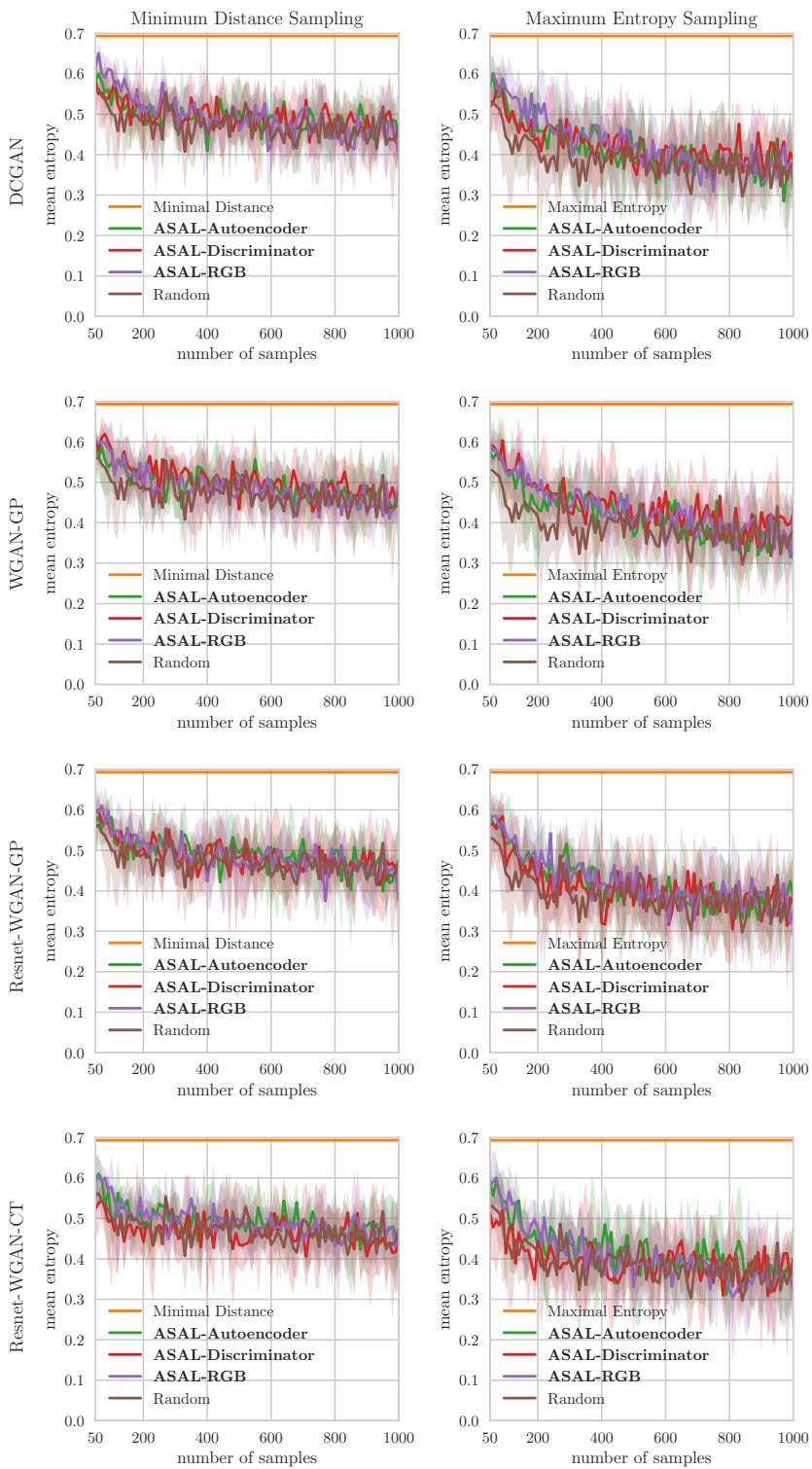

Figure 23: Average entropy of images that are selected and added to the training set for *CIFAR-10 - two classes* using different GANs. The mean entropy of the random sampling and the proposed method show hardly any difference. However, for maximum entropy sampling at least at the beginning ASAL selects images with higher entropy than random sampling.

## G    ADDITIONAL RESULTS: CIFAR - TEN CLASSES

For CIFAR-10, we do not indicate the true label distribution by a tick because the validation set contains the same number of samples for each class.

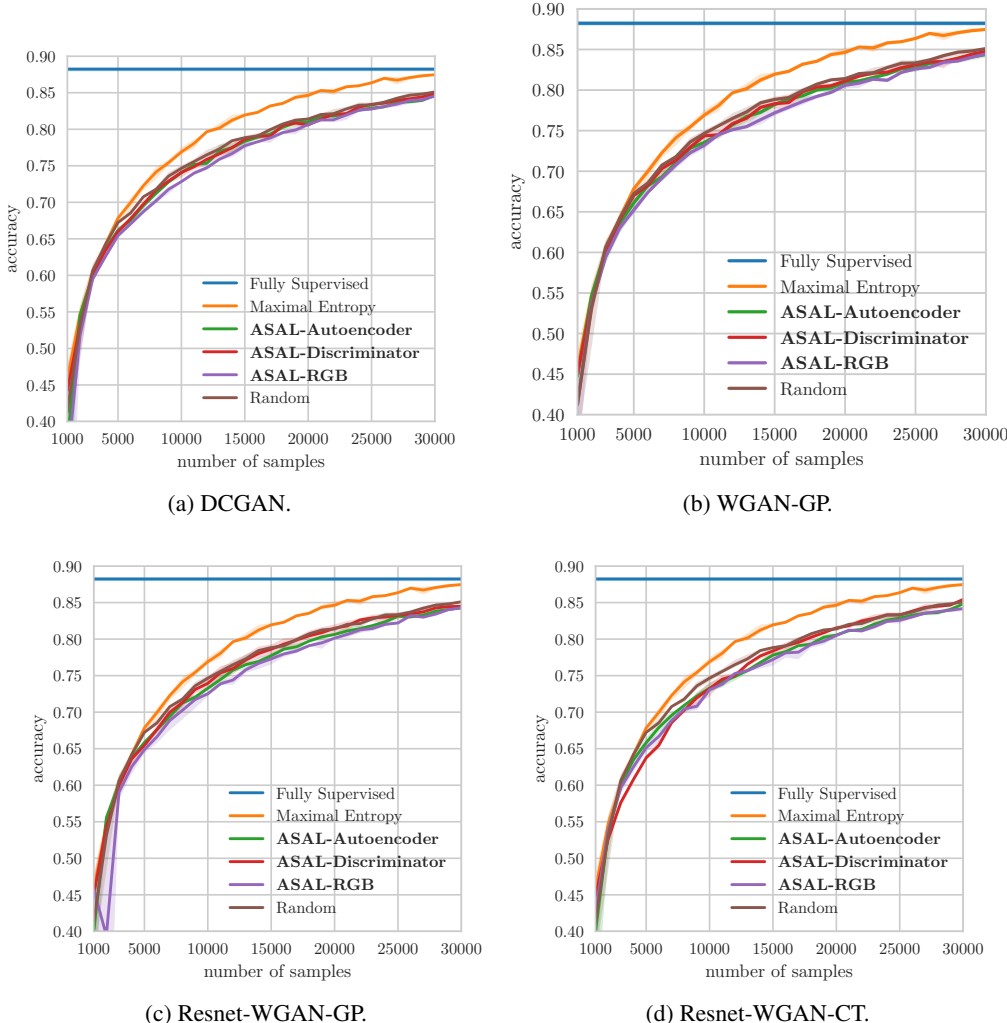

(a) DCGAN.

(b) WGAN-GP.

(c) Resnet-WGAN-GP.

(d) Resnet-WGAN-CT.

Figure 24: Validation accuracy on *CIFAR-10 - ten classes* of a fully supervised model, for random sampling, uncertainty sampling and different ASALs using different GANs. The proposed method performs slightly worse than random sampling independent of the sample matching of GAN.

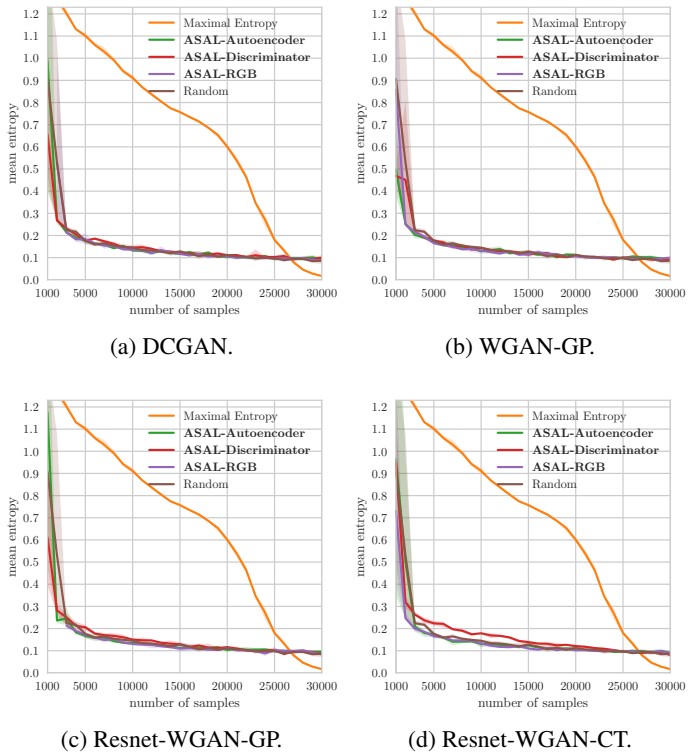

(a) DCGAN.

(b) WGAN-GP.

(c) Resnet-WGAN-GP.

(d) Resnet-WGAN-CT.

Figure 25: Average entropy of images that are selected and added to the training set for *CIFAR-10 - ten classes* using different GANs. There is hardly any difference for random sampling and ASAL in the entropy of newly added samples. Only at the beginning, random sampling retrieves samples with slightly higher entropy.

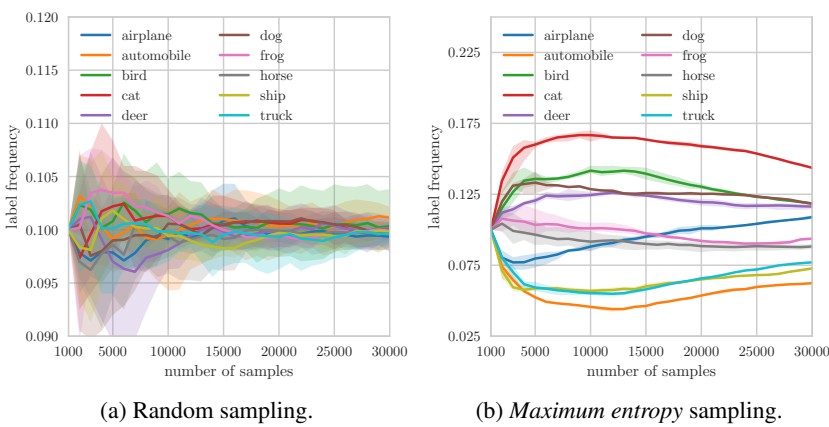

(a) Random sampling.

(b) *Maximum entropy* sampling.

Figure 26: Label distribution for uncertainty sampling using maximum entropy and random sampling for *CIFAR-10 - ten classes*. Random sampling converges to the true label distribution in the pool. Maximum entropy sampling selects most frequently cat, dog, bird, deer and least frequently automobile, ship, truck to exceed the classification quality of random sampling.

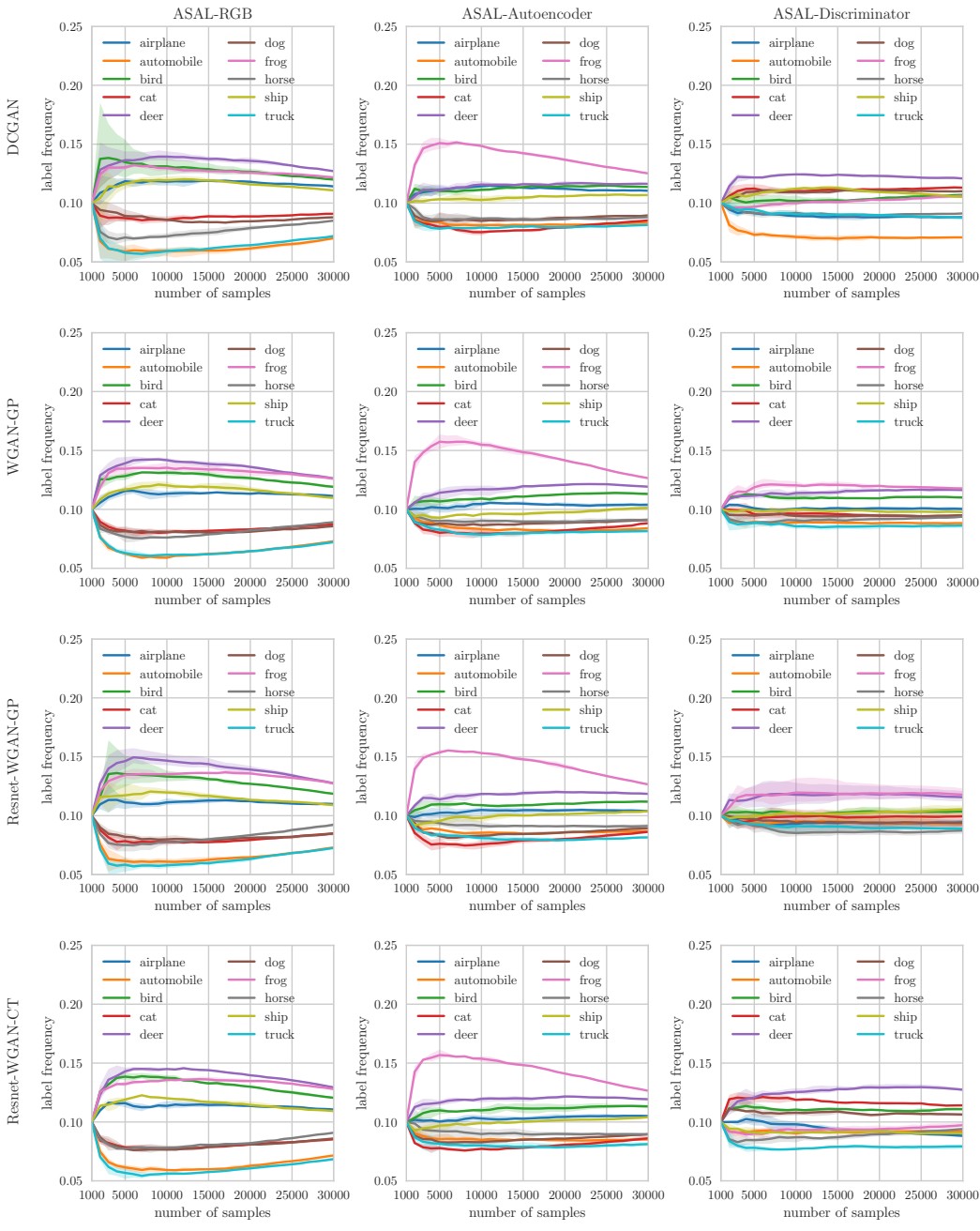

Figure 27: Label distribution for active learning using different matching strategies, uncertainty measures and GANs for *CIFAR-10 - ten classes*. Exactly the classes cat, dog that are most common in the training set of uncertainty sampling are less common in the data sets of most setups. Conversely, frog is for many setups the most common class but is not particularly frequent in the uncertainty sampling data set.

## H  MATCHING STRATEGY VISUALIZATION

Figs. 28, 29, 30, 31 show examples of generated images of the same active learning cycle and the corresponding matches. All images are generated using WGAN-GP and the maximum entropy score. The generated images are not manually annotated. The moderate quality of the generated CIFAR-10 images prevents confidently annotating the images. Instead, *n.a.* indicates that the manual annotation is missing.

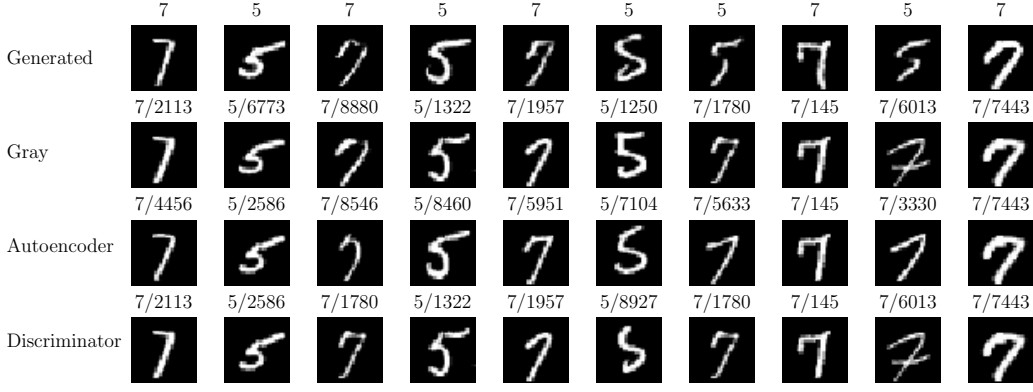

Figure 28: The first row shows synthetic digits and the other the closest samples from the pool using different features for comparison. The numbers above the image denote the label and image id.

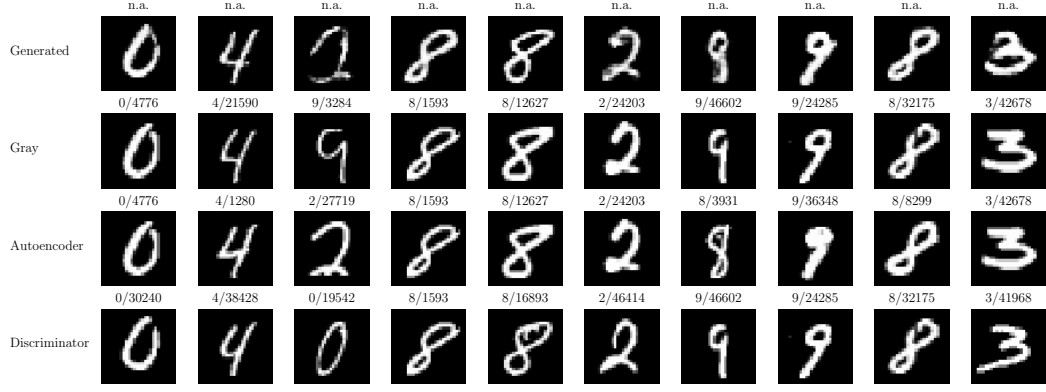

Figure 29: The rows show generated and matched images for *MNIST - ten classes* using WGAN-GP.

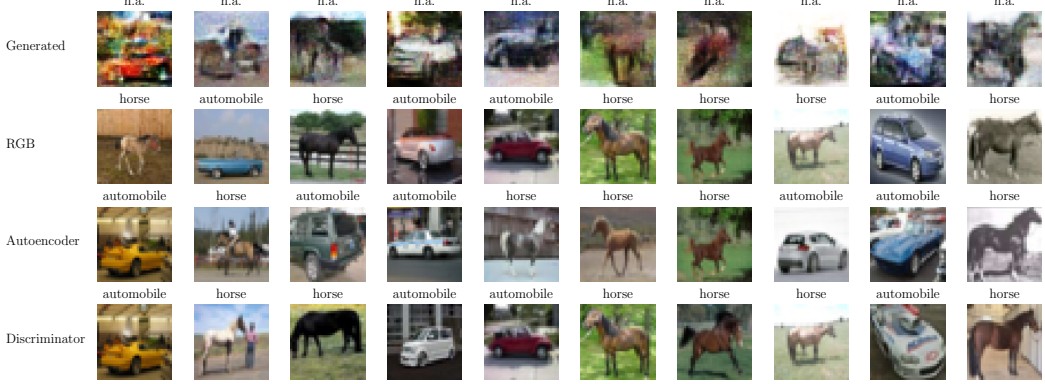

Figure 30: The rows show generated and matched images for *CIFAR-10 - two classes* using WGAN-GP. The images have a reasonable quality and all matching strategies retrieve images that are visually close or show the same class.

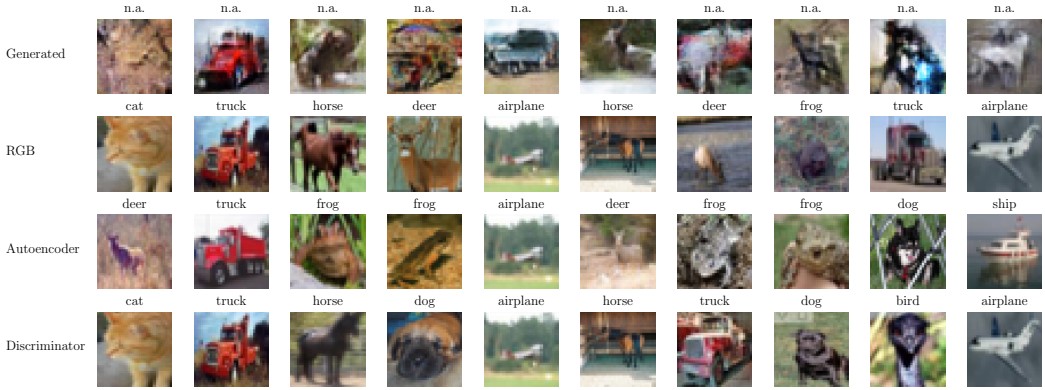

Figure 31: The rows show generated and matched images for *CIFAR-10 - ten classes* using WGAN-GP. Most of the generated images achieve only a moderate quality and even the closest samples from the pool have a high perceptual visual distance or assign images that show non matching classes, see last column where the images have a similar appearance but an appropriate label for the generated images would be horse but the selected samples show airplane and ship.

# I  GENERATED UNCERTAIN SAMPLES

To produce the images displayed in Figs. 32, 33, 34 and 35 we trained the classifier using the initial training set. Then we used maximum entropy sample generation to produce samples with a high entropy.

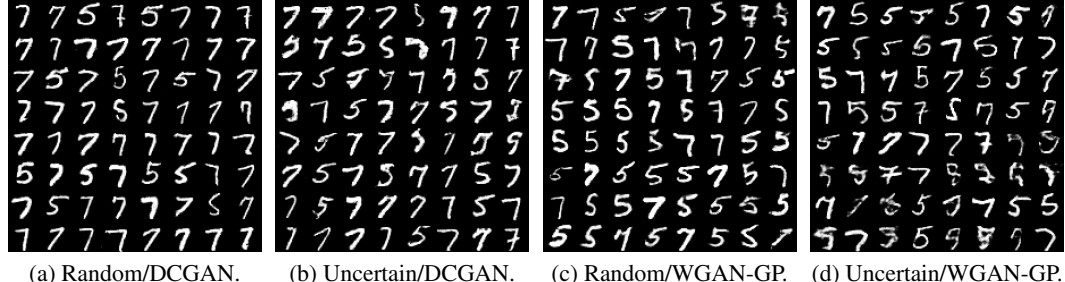

(a) Random/DCGAN.     (b) Uncertain/DCGAN.     (c) Random/WGAN-GP.     (d) Uncertain/WGAN-GP.

Figure 32: Comparison of random and uncertain *MNIST - two classes*. The samples are generated using different GANs. The *random* samples are visually more appealing and identifying the label is easier than for the *uncertain* samples. WGAN-GP generate images for both digits equally likely, whereas DCGAN most frequently generates images showing the digit 7.

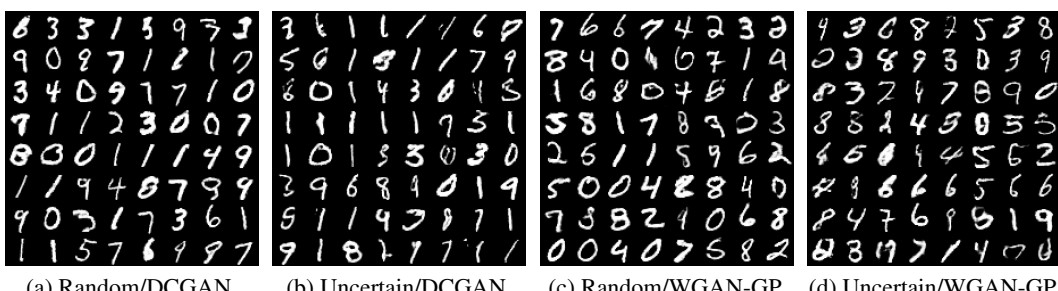

(a) Random/DCGAN.     (b) Uncertain/DCGAN.     (c) Random/WGAN-GP.     (d) Uncertain/WGAN-GP.

Figure 33: Comparison of random and uncertain *MNIST - ten classes* samples. The samples are generated using different GANs. The *random* samples are visually more appealing and identifying the label is easier than for the *uncertain* samples. WGAN-GP uniformly generates images for all digits, whereas DCGAN mainly generates images showing the digit 1.

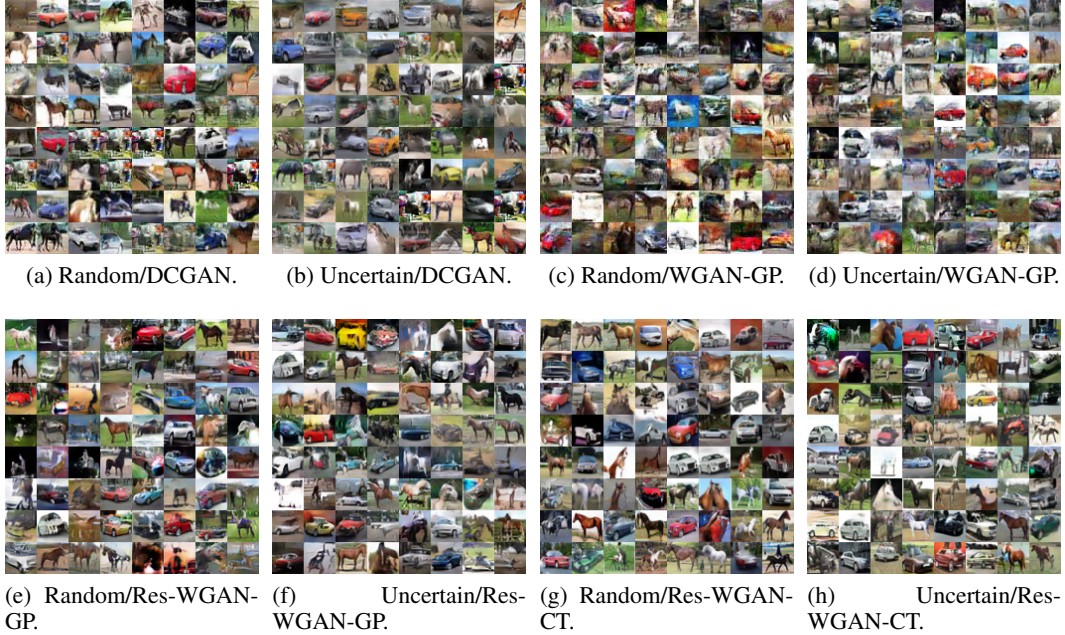

(a) Random/DCGAN.  (b) Uncertain/DCGAN.  (c) Random/WGAN-GP.  (d) Uncertain/WGAN-GP.

(e) Random/Res-WGAN-GP.  (f) Uncertain/Res-WGAN-GP.  (g) Random/Res-WGAN-CT.  (h) Uncertain/Res-WGAN-CT.

Figure 34: Comparison of random and uncertain samples for *CIFAR-10 - two classes* using maximum entropy. The samples are generated using different GANs. The residual GANs (bottom row) produce more visually appealing samples than the other GANs. For most of these images it would be possible to identify whether the image shows a horse or automobile.

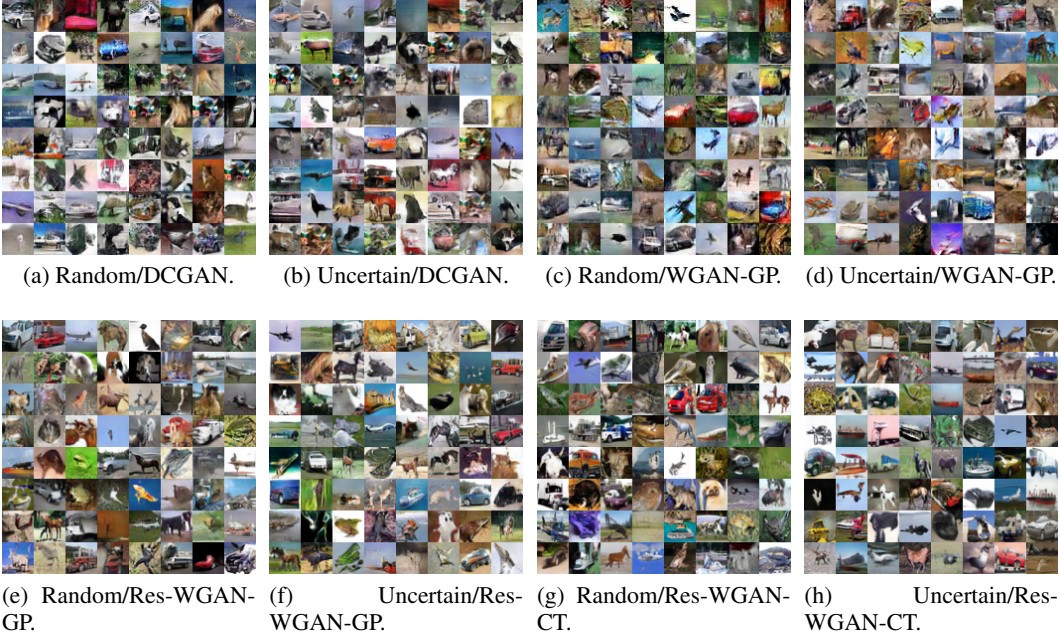

(a) Random/DCGAN.  (b) Uncertain/DCGAN.  (c) Random/WGAN-GP.  (d) Uncertain/WGAN-GP.

(e) Random/Res-WGAN-GP.  (f) Uncertain/Res-WGAN-GP.  (g) Random/Res-WGAN-CT.  (h) Uncertain/Res-WGAN-CT.

Figure 35: Comparison of random and uncertain samples for *CIFAR-10 - ten classes* using maximum entropy. The samples are generated using different GANs. The residual GANs (bottom row) produce more visually appealing samples than the other GANs. Although, the quality of the *random* images is higher than of the *uncertain* images, annotating with high confidence is still very difficult.

