# OpenReview forum: "Adversarial Sampling for Active Learning"
_ICLR.cc/2019/Conference_

### Official Review · AnonReviewer3 · 2018-10-31
**GAN-based active learning for query synthesis**

**Rating:** 5
**Confidence:** 5

**Review:**

This paper proposed a query-synthesis-based active learning algorithm that uses GAN to generate high entropy sample; instead of annotating the synthesized sample, the paper proposed to find the most similar unlabeled data from the pool via nearest neighbor search, with the latter is the main contribution of the paper.

Pros:
(1)	the paper is well written and easy to follow;
(2)	evaluations look reasonable and fair

Cons:
(1)	The idea of using GAN for active query synthesis isn’t new. As the authors pointed out, this idea is mainly from GAAL (Zhu & Bento 2017). The main difference is sample matching that searches the nearest neighbor from pool and add the real unlabeled data for AL. So the novelty of the paper isn’t significant.
(2)	In terms of accuracy comparison, on Cifar-10-ten classes experiments, all ASAL variants have similar accuracies as random sampling, while traditional pool-based max-entropy clearly works much better. Although the former is much faster (O(1) vs. O(N)), this benefit is mainly due to GAAL (Zhu & Bento 2017).

The paper provides additional evidence showing that GAN-based active learning might be an interesting research direction for active query synthesis. However, given the reasons above, particularly novelty, I think the authors might need to additional work to improve the method.

---

> ### Author Response · Authors · 2018-11-25
> **Difference between ASAL and GAAL**
>
> Thanks for the review. Although you are very confident about your evaluation we feel that we need to clarify the differences to previous methods such as GAAL, as sample matching is not the only but the most apparent difference among many other. Furthermore, we use sample generation in a very different context. Our goal is to approximate uncertainty sampling with sub-linear run-time complexity. Whereas, the goal of GAAL is reaching better classification performance with fewer samples than traditional methods.
> Furthermore, we agree that the proposed ingredients are rather simple but crucial to enable for the first time GAN based uncertainty sampling that is not only efficient but also effective. Furthermore, we run ASAL on three additional benchmarks (see Appendix A) where it consistently outperforms random sampling and selects new samples up to 27 times faster than classical uncertainty sampling.
> Note, that we also included the results of GAAL on CIFAR-10-two classes (see Fig. 4b) and observe that it performs much worse than all other methods and thus highlights again the effectiveness of out method. The reason why ASAL is not effective on CIFAR-10 - ten classes is mainly due to the rather small data set compared to its complexity.
>
> We see the following contributions compared to GAAL.
> - ASAL and GAAL correspond to completely different classes of active learning (pool based vs query synthesis)
> - GAAL requires a fixed feature space and uses linear SVMs. ASAL allows to use CNNs with varying feature space.
> - GAAL uses the minimum distance criterion limited to two classes. ASAL uses the maximum entropy criterion
>    extended to multiple classes.
> - GAAL requires to annotate synthetic samples that are difficult to label. ASAL selects real samples that are
>   easier to label and can be reused.
> - GAAL performs worse than random sampling and suffers from sample bias. ASAL performs better than
>    random sampling and is robust to sample bias.
>
> To summarise, we only use the sample generation idea of Zhu & Bento but developed additional components that are crucial to enable GAN base active learning and to outperform GAAL. Supported by the experiments in the main paper and the additional ones on CelebA, we hope to convince the reviewer that developing simple components that make GAN based active learning work for the first time, with sublinear run-time complexity and for CNNs, are more than sufficient contributions.

---

### Official Review · AnonReviewer1 · 2018-11-03
**Nice Idea but Weak Experiments**

**Rating:** 5
**Confidence:** 4

**Review:**

The paper presents a pool-based active learning method that achieves sub-linear
runtime complexity while generating high-entropy samples, as opposed to linear
complexity of more traditional uncertainty sampling (i.e., max-entropy) methods.
This is achieved by using a generative adversarial network (GAN) to generate
high-entropy samples that are then used by a nearest neighbor method to pick
samples from a pool, that are closest to the generated samples. The sub-linear
complexity is achieved through the use of a k-d tree, combined with the fact
that similarity is computed on the feature space and samples can thus be indexed
once (as the feature space does not change while training).

The proposed idea builds on top of previously published work on Generative
Adversarial Active Learning (GAAL). The main difference is the added nearest
neighbor component, as GAAL is directly using the generated examples, thus
achieving constant runtime complexity, rather than sub-linear.

I like the overall direction and the idea of being able to perform uncertainty
sampling in sub-linear time. The approach is interesting. However, the results
presented in the paper are not strong and I do not see whether or not I should
be using this method over uncertainty sampling. Most importantly, the results
are strongest only for the MNIST experiments, which are over a small dataset.
Given that the method is motivated by the scalability argument, I would like to
see at least one large scale experiment where it performs well, and more
specifically, outperform random sampling. Also, I would really like to see a
more principled and thorough experimental investigation with justifications for
the configurations used and with more comparisons to alternative approaches,
such as GAAL, which has constant complexity.

I believe that it would be better for your paper if you work a bit more on the
experimental evaluation and submit a revised version at a later deadline.

== Background and Method ==

The background and method sections are clear and easy to follow. One improvement
I can see is making figure 1 more clear, by maybe explicitly stating in the
figure what "G" and "F" are. One more point of interest is that the way you
perform sample matching makes some smoothness assumption about the feature
space as related to the classifier uncertainty. I perceive this as a smoothness
assumption on the decision boundary of the classifier and I do not know how true
is may be for deep neural networks, but I can see how it may be true for
logistic regression models and support vector machines (SVMs), depending on the
kernel used. I believe that this point and main assumption may be worth further
discussion, given that it is also about the main difference your method has with
respect to GAAL.

I do not have any other major comments for these sections as my main concerns
are about the experiments section.

== Experiments ==

In the experiments, it would be very useful to have plots against execution
time, given that the main motivation for this method is scalability. For
example, the method outperforms random sampling for small datasets, based on
number of samples, but what happens when you look at execution time? Given that
random sampling is very cheap, I imagine that it probably does better. Also, as
mentioned earlier, I would like to see at least one experiment using a big
dataset, where the method outperforms random sampling, as I am not currently
convinced of its usefulness.

Also, you present a lot of results and list observations but I felt there was
not much discussion as to why you observe/obtain some of the results. Given that
your method is not working very well for CIFAR, I would like to see a more
thorough investigation as to why that may be the case. This investigation could
conclude with some "tips" on when it may be a good idea to use your method over
GAAL, or uncertainty sampling, for example.

Regarding the experimental setup, I find lots of configuration choices very
arbitrary and have difficulty understanding how they were chosen. For example:

  - For the two-class MNIST you use classes "5" and "7" and for the two-class
    CIFAR you use classes "automobile" and "horse". Why is that? How did you
    pick the two classes to use in each case? Do the results match for other
    class pairs?
  - "learning rate of 0.01 that we decay by a factor of 10 at the 130th and
    140th epochs" -- end of page 6
  - "In contrast to the previous experiments we use a residual Wasserstein GAN
    with gradient penalty and soft consistency term" -- page 7 -- why do you
    make that change?

Questions:
  - Why do you think using Wasserstein GANs perform better than using DCGANs? -- section 5.3.1
  - Why not compare to GAAL in all of figures 3 and 4?
  - How/why were the number of samples you start with and sample in each round,
    chosen? Do you observe any difference if you increase/decrease the number of
    samples sampled in each round or if you start with fewer samples?
  - How/why were these model architectures chosen?

---

> ### Author Response · Authors · 2018-11-25
> **New experiments available on a large data set where ASAL outperforms random sampling.**
>
> Thanks for your thorough review. We appreciate the interest in our method and will first try to address the general question or doubts before going into details of the short questions.
>
> First, we want to point out that we created three new benchmarks on CelebA, that contains more than 200k samples (see Appendix A). We show the efficiency and effectiveness of ASAL and outperform random sampling on all three benchmarks and select new samples up to 27 times faster than maximal entropy sampling. Furthermore, we took the time to manually label several thousand synthetic images to add the performance of GAAL for CIFAR-10 - two classes, see Fig. 4b (we stopped annotating once it was clear that its performance is way below random sampling).
>
> ASAL should be used for huge data sets where even running algorithms with a linear run-time complexity is too expensive and a better performance than random sampling is desired. To the best of our knowledge, ASAL is the only uncertainty based active learning method with sub-linear run-time complexity that supports CNNs and achieves better classification quality than random sampling. Using ASAL or random sampling instead of GAAL is always beneficial for any setup, following the conclusion of Zhu & Bengio and considering our experiments using GAAL.
>
> We compare the algorithms when they uses the same amount of labelled samples for training. If we understand your suggestions correctly, you suggest to compare them in terms of run-time, i.e. random sampling quickly selects new samples thus there is more time to annotate samples compared to ASAL or uncertainty sampling.
> Note, that for 160k samples, random sampling takes a few milliseconds, ASAL 1.15s and uncertainty sampling 32s per active learning cycle. For 16M samples ASAL takes less than a minute and uncertainty sampling more than 50 min.Thus, the savings of random sampling compared to ASAL does not allow to label many additional samples, but the savings compared to uncertainty sampling would. However, labelling time might vary heavily between different samples or data sets. Thus, we feel that comparing active learning algorithms in the traditional way (same number of labelled samples) lead to a fairer comparison even though ASAL would look even more promising using the proposed idea.
>
> Indeed, when fixing the features and training a linear SVM on top there will be some uncertainty smoothness in the features space (samples closer to the separating hyperplane have higher uncertainty and samples further away have a smaller uncertainty). Thus, samples in a local vicinity in the feature space have a similar uncertainty.
> However, ASAL fixes the “matching feature space” first and allows to update the features of the CNN. Thus, using the features of the last layer of the CNN, nearby samples in this feature space will have a similar uncertainty. However, two samples that are close in the “CNN feature space” might not be close in the “matching feature space” and vice versa.
> Fortunately, we  do not require the last property but need a feature space where locally the uncertainty of samples is similar and a model that generates uncertain samples that follow the true data distribution. Using such a model ensures that the uncertain samples are realistic and will be uncertain because of their visual content rather than small invisible adversarial components. This property is crucial because we choose the “matching feature space”  based on the main visual features present in the images. Therefore, we would match a true image with its adversarial perturbed sample (different entropy of both images). Conversely, a sample that is uncertain because of its content (person wearing glasses) will have nearby samples in the “matching feature space” that have similar visual content.
> Thus, the proposed matching depends on the quality of the GAN, how densely the data set covers the feature space and how characteristic these features are for the data set and the classification task. Thus, ASAL is fast on huge data sets and due to the denser coverage of larger data sets the better agreement of the matches leads to better performance (at the limit, the feature space will be fully covered and we will find a fully agreeing match for each generated uncertain sample). Although we have no theoretical guarantee, our experiments on CelebA and MNIST show that for large dataset (depending on the complexity of the samples) ASAL works. Note, that MNIST is rather small but 5k different samples for a single grayscale digit cover almost all variant. Conversely, 5k different images of more complex objects (such as car) are far from covering all possible variants.
>
> To summarise, as requested, we showed that ASAL outperforms random sampling on three new and larger data set. We compare to GAAL on another data set and give some explanation why and when ASAL works. We hope that these additional experiments and explanations convince the reviewer of the contributions of this work.

---

> > ### Author Response · Authors · 2018-11-25
> > **Answer of detailed questions**
> >
> > “For the two-class MNIST you use classes "5" and "7" and for the two-class  CIFAR you use classes "automobile" and "horse". Why is that? How did you pick the two classes to use in each case? Do the results match for other class pairs?”
> >
> > We use this particular subsets of MNIST and CIFAR-10 because we want to compare to Zhu & Bengio that proposed this data sets in their GAAL paper. We did not exhaustively check all the possible combination but validated on the full data set.
> >
> > “How/why were the number of samples you start with and sample in each round chosen? Do you observe any difference if you increase/decrease the number of samples sampled in each round or if you start with fewer samples?”
> >
> > Again we use the same setup for the binary experiments as proposed by Zhu & Bengio in their GAAL paper (25 initial samples per class and adding 10 each cycle). For MNIST with ten classes, the initial data set contains 10 samples and we add 50 samples each cycle.
> > Samples that are selected within the sample cycle are correlated thus reducing the number of selected samples per cycle reduces correlation of the full active learning data set for a given budget. Thus, a smaller sampling size leads to better accuracy but substantially increases training time. For CIFAR-10 we start with 100 samples for each class. Typically we reach the best performance if we actively select as many samples as possible, thus keeping the initial data set small. However, ASAL and uncertainty sampling only work if the uncertainty estimation is reliable. For example if the model performs very poorly and assigns each sample with high probability the wrong label no sample has a high entropy. Thus, we observed that 1000 initial samples are enough for the data set and architecture. Furthermore, we add 1000 data sample in each AL cycle. We use that many samples because we observed that adding fewer samples does not improve the performance significantly but takes much longer to train for the same labelling budget. Again, adding more samples at once comes with the risk that we add to many samples sharing the same properties.
> >
> > "learning rate of 0.01 that we decay by a factor of 10 at the 130th and 140th epochs" -- end of page 6
> >
> > We use the architecture and training policy proposed by Springenberg et al. which achieves state of the art results on CIFAR-10. However, we observed that the model trained on a smaller training set converges faster. Hence we use 150 training epochs.
> >
> > “How/why were these model architectures chosen?”
> >
> > As already mentioned we use either the models proposed by Zhu & Bengio or models that are well know in the literature for the corresponding data sets or because they achieve state of the art performance (LeNet, All-Conv-C, Wasserstein GAN with gradient penalty and/or consistency term).
> >
> > "In contrast to the previous experiments we use a residual Wasserstein GAN  with gradient penalty and soft consistency term" -- page 7 -- why do you  make that change?
> >
> > We observed that the a higher quality of the GAN has a positive impact on ASAL. Therefore, we use a residual GAN with gradient penalty and consistency term, that was published this year at ICLR, and achieves a state of the art inception score. Nonetheless, we performed all the experiments with different GANs, see Figs. 22 and 24 in the Appendix.
> >
> > “Why do you think using Wasserstein GANs perform better than using DCGANs?”
> >
> > Gulrajani et al. observed that the Wasserstein GAN with gradient penalty leads to higher quality GAN than training a DCGAN. Furthermore, we observed that the DCGAN trained on MNIST produces 7’s much more frequently than 5’s compared to the distribution in the training set. Conversely, the distribution obtained from the Wasserstein GAN matches the data distribution. See, Figs 11 and 32 in the Appendix. Fig 32a shows randomly selected samples from the DCGAN although the digits 5 and 7 are almost equally likely in the data set, three out of eight rows show only 7’s.
> >
> > “Why not compare to GAAL in all of figures 3 and 4?”
> >
> > As already mentioned we updated Fig 4b such that it includes GAAL now. Furthermore, we discussed that GAAL cannot be used with CNNs and extending the work of Zhu & Bengio would be required for multi-class problems. Note, that implementing GAAL for the two benchmarks would require manually labelling ten-thousands of synthetic samples that are difficult to annotate and cannot be reused for any other task. Doing this for multiple random seeds would be even more expensive. Furthermore, Zhu & Bengio conclude that GAAL performs worse than random sampling therefore we felt that further comparing to GAAL is pointless.

---

### Official Review · AnonReviewer2 · 2018-11-11
**Nice idea for important problem, but requires validation on algorithm speed.**

**Rating:** 6
**Confidence:** 2

**Review:**

This paper proposes adversarial sampling for pool-based active learning, which is a sublinear-time algorithm based on 1) generating “uncertain” synthetic examples and 2) using the generated example to find “uncertain” real examples from the pool. I liked the whole idea of developing a faster algorithm for active learning based on the nearest neighborhood method. However, my only & major concern is that one has to train GANs before the active learning process, which might cost more than the whole active learning process.

Pros:
- This paper tackles the important problem of reducing the time complexity needed for active learning with respect to the pool size. I think this is a very important problem that is necessary to be addressed for the application of active learning.
-The paper is well written and easy to understand.
-I think the overall idea is novel and useful even though it is very simple. I think this work has a very promising potential to be a building block for future works of fast active learning.

Cons:
-There is no theoretical guarantee on the "uncertainty" of obtained real examples.
-The main contribution of this algorithm is computational complexity, but I am not very persuaded by the idea of using the GAN in order to produce a sublinear (faster) time algorithm for active learning, since training the GAN may sometimes take more time that the whole active learning process. Explicitly describing situations where the proposed method is useful seems necessary. I would expect the proposed algorithm to be beneficial when there is a lot of queries asked and answered, but this seems unlikely to happen in real situations.
-Empirical evaluation is weak, since the algorithm only outperforms the random sampling of queries. Especially, given that sublinear nature of the algorithm is the main strength of the paper, it would have been meaningful to evaluate the actual time spent for the whole learning process including the training of GANs. Especially, one could also speed-up max entropy criterion by first sampling subset of data-points from the pool and evaluating upon them.

---

> ### Author Response · Authors · 2018-11-25
> **New experiments available in the appendix that report the requested timings**
>
> Thanks for the review and for the precise summary of the main idea proposed in our paper. We updated Fig. 4b such that it now contains GAAL for CIFAR-10 two classes. This experiment shows, that our simple and efficient matching strategy is crucial to successfully run GAN based active learning.
> Furthermore, to address the questions of the reviewers we run additional experiments on a new and larger benchmark (CelebA) and show that ASAL works reliably. We report the timings for one active learning cycle and the whole training process when including the training time of the GAN (see Appendix A in the revised paper).
>
> You propose to compare our method to other algorithms in active learning. However, our intention is not to propose a new active learning algorithm that outperforms existing methods but rather to approximate classical uncertainty sampling with a smaller run-time complexity. Therefore, we report random sampling as a baseline and maximum entropy sampling as the upper bound. As explained in the related work section, to the best of our knowledge, there is only one other pool-based active learning algorithm (Jain et al.) that tackles the same problem. However, their algorithm cannot be used as soon as the features change during active learning as required for CNNs. Thus, a fair comparison that uses the same models for both algorithms is impossible by design. Note, that GAAL has similar limitations but because we extend the sample generation idea we felt that comparing to GAAL would be useful. Thus, we run experiments only using an SVM. So far we run GAAL on MNIST but we added experiments using GAAL for CIFAR-10 - two classes. We see that GAAL performs clearly worse than all other methods (We stopped after 400 samples as the trend is already clear and annotating more samples would just be a waste of time).
>
> We share your opinion that it would be useful to report the timings of sample selection and the whole active learning process when including the pre-processing times even though ASAL has the better run-time complexity. Thus, we added a new set of experiments on CelebA, see Appendix A. We see, that there exists always a transition point, where ASAL gets more efficient than uncertainty sampling when respecting the pre-processing time. Furthermore, on this data set, ASAL selects samples up to 27 times faster than maximal entropy sampling. Note, that the conducted experiments in the paper are validations of ASAL on well known data sets. However, the data sets are not large enough to amortise the training of the GAN. Considering the high diversity of possible face images and that many face attributes are underrepresented in CelebA we see this data set still as a rather small data set.Thus, for real world applications constructing a much larger data set seems reasonable. We measured the timings of all components for different data set sizes based on CelebA and computed the corresponding run times rather than performing the full experiments because running uncertainty sampling would be to expensive for the used settings.
>
> Although using the full pool is clearly beneficial, we agree that using the known approach of randomly subsampling the pool first and then running uncertainty sampling over the selected samples speeds up the runtime. However, for huge data sets a high subsampling factor is required to achieve affordable sampling times. With such a high subsampling factor it is very likely that we miss many uncertain samples when training converges. Thus, an adaptive subsampling factor might be required for effective active learning that depends on the model and data set. Thus, this approach would still lead to a linear run-time complexity but is faster in practice than searching the full pool.
> Furthermore, ASAL allows to generate more than the required samples and allows to scan the matches of these samples for the most uncertain ones. As we already know that ASAL performs better than random sampling it is very likely that using ASAL instead of random sampling to select subsets will work even better and requires less scanning. We felt that reporting the performance of the raw methods leads to a more transparent and fairer comparison.
>
> To summarise, we showed that there are settings where ASAL outperforms uncertainty sampling even when taking training the GAN into account. Furthermore, we showed on a new data set with three benchmarks the effectiveness of ASAL. We hope that these additional experiments convince the reviewer of the contributions of this work.

---

### Meta-Review · Area_Chair1 · 2018-12-12
**Limited experimental results**

**Confidence:** 4
**Recommendation:** Reject

**Metareview:**

The paper proposes adversarial sampling for pool-based active learning.

The reviewers and AC note the critical potential weaknesses on experimental results: it is far from being surprising the proposed method is better than random sampling. Ideally, one has to reduce the complexity under keeping the state-of-art performance. Otherwise, it is hard to claim the proposed method is fundamentally better than prior ones, although their targets might be different.

AC thinks the proposed method has potential and is interesting, but decided that the authors need more works to publish.